# Few regulatory metabolites coordinate expression of central metabolic genes in *Escherichia coli*

Karl Kochanowski[1,2,†] ID, Luca Gerosa[1,2,†] ID, Simon F Brunner[1] ID, Dimitris Christodoulou[1,2], Yaroslav V Nikolaev[3] ID & Uwe Sauer[1,*] ID

## Abstract

Transcription networks consist of hundreds of transcription factors with thousands of often overlapping target genes. While we can reliably measure gene expression changes, we still understand relatively little why expression changes the way it does. How does a coordinated response emerge in such complex networks and how many input signals are necessary to achieve it? Here, we unravel the regulatory program of gene expression in *Escherichia coli* central carbon metabolism with more than 30 known transcription factors. Using a library of fluorescent transcriptional reporters, we comprehensively quantify the activity of central metabolic promoters in 26 environmental conditions. The expression patterns were dominated by growth rate-dependent global regulation for most central metabolic promoters in concert with highly condition-specific activation for only few promoters. Using an approximate mathematical description of promoter activity, we dissect the contribution of global and specific transcriptional regulation. About 70% of the total variance in promoter activity across conditions was explained by global transcriptional regulation. Correlating the remaining specific transcriptional regulation of each promoter with the cell's metabolome response across the same conditions identified potential regulatory metabolites. Remarkably, cyclic AMP, fructose-1,6-bisphosphate, and fructose-1-phosphate alone explained most of the specific transcriptional regulation through their interaction with the two major transcription factors Crp and Cra. Thus, a surprisingly simple regulatory program that relies on global transcriptional regulation and input from few intracellular metabolites appears to be sufficient to coordinate *E. coli* central metabolism and explain about 90% of the experimentally observed transcription changes in 100 genes.

**Keywords** metabolism; modeling; network inference; transcription factors; transcriptional regulation
**Subject Categories** Genome-Scale & Integrative Biology; Metabolism; Transcription
**Mol Syst Biol. (2017) 13: 903**

## Introduction

For most microbes, growth-sustaining environments encompass a wide range of nutritional conditions (Bochner *et al*, 2001; Nichols *et al*, 2011; Orth *et al*, 2011). Coping with such diverse environments requires coordination of metabolism to provide biomass precursors, redox factors, and energy at appropriate stoichiometries (Chubukov *et al*, 2014). Transcriptional regulation plays a key role in adaptation, such as the induction of uptake and utilization pathways upon nutrient availability (Kaplan *et al*, 2008; Aidelberg *et al*, 2014; Chubukov *et al*, 2014). Beyond such relatively simple control of particular pathways, transcriptional regulation is typically much more complex, where even subtle environmental changes alter expression of hundreds of metabolic genes (Kao *et al*, 2004; Liu *et al*, 2005; Jozefczuk *et al*, 2010; Costenoble *et al*, 2011; Buescher *et al*, 2012; Nicolas *et al*, 2012). How are such global transcriptional responses achieved, and are all of these changes actually required for a given adaptation?

Most work focused on the role of transcription factors, and efforts in mapping out their targets have cumulated in highly overlapping and dense transcriptional regulatory networks, comprising major (with hundreds of target genes) and minor (with few target genes) transcription factors (Janga *et al*, 2007; Goelzer *et al*, 2008; Martínez-Antonio *et al*, 2008; Seshasayee *et al*, 2009; Cho *et al*, 2012; Salgado *et al*, 2013; Teixeira *et al*, 2014). To exert their regulatory function, transcription factors, in turn, need to receive signals. These signals may be relayed through two-component systems where detection of internal or external signals by a designated receptor protein is coupled to the phosphorylation of the respective transcription factor (Laub & Goulian, 2007). Alternatively, transcription factor activity may also respond directly to levels of regulatory metabolites. Examples of such regulatory metabolites include amino acids that frequently regulate expression of their own biosynthesis pathway, intermediates of nutrient utilization pathways such as glycerol phosphate, and central metabolites such as pyruvate or fructose-1,6-bisphosphate (Chubukov *et al*, 2014). This "one-component" type of regulation is prevalent in bacteria (Ulrich *et al*, 2005), allowing

1 Institute of Molecular Systems Biology, ETH Zurich, Zurich, Switzerland
2 Life Science Zurich PhD Program on Systems Biology, Zurich, Switzerland
3 Institute of Molecular Biology & Biophysics, ETH Zurich, Zurich, Switzerland
*Corresponding author. Tel: +41 44 633 36 72; E-mail: sauer@imsb.biol.ethz.ch
†These authors contributed equally to this work

them to match the transcriptional response to their current metabolic state. Thus, a cell's transcriptional response is the result of the complex interplay of many transcription factors whose activity is adjusted to the environment by their respective regulatory signals (Janga *et al*, 2007; Martínez-Antonio *et al*, 2008; Seshasayee *et al*, 2009).

Understanding how a cell's transcriptional response emerges from this interplay of complex regulatory networks and signals is a daunting task, and two key limitations have hampered our ability to study such regulatory networks in a quantitative manner. The first limitation stems from the realization that the activity of a promoter is determined not only by the network of transcription factors (termed "specific transcriptional regulation"), but also by the global physiology of the cell (Liang *et al*, 1999; Dennis *et al*, 2004; Zaslaver *et al*, 2009; Scott *et al*, 2010; Berthoumieux *et al*, 2013; Gerosa *et al*, 2013; Keren *et al*, 2013; Shahrezaei & Marguerat, 2015; Barenholz *et al*, 2016; termed "global regulation"), for example, through growth rate-dependent changes in free RNA polymerase availability or ribosome abundance (Bremer & Dennis, 1996; Klumpp & Hwa, 2008; Scott *et al*, 2010; Borkowski *et al*, 2016). This global regulation adds another layer of complexity and is responsible for a large fraction of the gene expression response in bacteria and eukaryotes (Keren *et al*, 2013). To understand this response, we therefore need to robustly dissect the contribution of global and specific transcriptional regulation for large-scale networks across diverse conditions. First attempts have been made but were restricted to detailed dynamic analysis of few promoters (Berthoumieux *et al*, 2013; Gerosa *et al*, 2013), or large-scale analysis for few conditions (Zaslaver *et al*, 2009; Keren *et al*, 2013). The second limitation is the lack of methods to systematically identify signals that regulate transcription factor activity. In particular, transcription factor–metabolite interactions are—as all protein–metabolite interactions (McFedries *et al*, 2013)—notoriously difficult to detect due to their non-covalent nature (Kochanowski *et al*, 2015). Consequently, such interactions have largely been identified through extensive biochemical analyses or serendipity on a case-by-case basis. Even for well-studied organisms such as *Escherichia coli*, regulatory signals for the majority of transcription factors have remained elusive (Babu *et al*, 2003). Efforts to infer such interactions directly from *in vivo* experimental data have been restricted to few case studies, without accounting for the confounding influence of global transcriptional regulation (Cakir *et al*, 2006; Bradley *et al*, 2009).

In this work, we aimed to overcome these limitations to unravel the transcriptional regulatory program of *E. coli* central carbon metabolism, whose extensive transcription factor network is topologically well characterized (Salgado *et al*, 2013) and responds to a wide range of environmental perturbations. Using a library of fluorescent transcriptional reporters, we comprehensively quantify the activity of central metabolic promoters under a wide range of environmental conditions. By establishing a mathematical model, we decompose each promoter's global and specific transcriptional regulation and systematically infer potential metabolite regulators of gene expression by correlation. A surprisingly simple regulatory program based on two transcription factors and very few intracellular regulatory metabolites is sufficient to coordinate transcription of *E. coli* central metabolism under a large number of conditions.

# Results

## Quantifying gene expression of *E. coli* using transcriptional fluorescent reporters

To unravel the transcriptional regulatory program governing *E. coli*'s central metabolism, we used a library of fluorescent transcriptional reporter plasmids (Zaslaver *et al*, 2006) and expanded it with 28 additional promoters to cover over 90% of the 100+ genes in central carbon metabolism. We further included five previously characterized synthetic constitutive promoters (Gerosa *et al*, 2013) that are only affected by global transcriptional regulation, hence allow to study the impact of global regulation in isolation (Table EV1 for full list of promoters). In total, we determined the activity of 95 promoters during steady-state growth (from time course measurements, see Appendix Fig S1) under 26 diverse conditions—including different carbon source and amino acid supplementation, as well as sub-lethal ribosome inhibition by chloramphenicol, which directly affects global regulation. Under these conditions, we achieved steady-state growth rates between 0.1 and 1.5/h (see Table EV2 for full list of conditions). Despite promoter activities spanning several orders of magnitude, there were no adverse effects of GFP expression on growth rate (Appendix Fig S2). Day-to-day comparison demonstrated reproducible promoter activity measurements during exponential growth within 10–20% variation, which is comparable to previous studies (Zaslaver *et al*, 2009; Keren *et al*, 2013; Appendix Fig S3). Thirty-one of the tested promoters, predominantly those of minor central metabolic isoenzymes, were inactive under all tested conditions and were therefore discarded for subsequent analyses.

The remaining 64 promoters showed few distinct patterns across conditions (Fig 1 and Appendix Fig S4, data in Table EV3). A small subset of promoters that were mostly part of carbon utilization pathways showed highly condition-specific activation in one or two of the tested conditions (e.g. *fruB*). Approximately 15% of the tested promoters, mostly consisting of TCA cycle promoters (e.g. *sdhC*), were activated on carbon sources supporting growth rates below 0.8/h, but were not activated by chloramphenicol treatment that led to similarly low growth rates. The activity of the majority of tested promoters, however, was consistently related to the growth rate supported by each condition (as confirmed by hierarchical clustering, Appendix Fig S5). Notably, all tested synthetic constitutive promoters, which are only affected by global regulation, fell into the last category.

## Dissecting global and specific transcriptional regulation in central carbon metabolism

Based on these data, we hypothesized that growth-dependent global regulation has a strong impact on central metabolic promoter activity (see also Appendix Fig S6). Next, we established a computational method to dissect the measured promoter activity of each promoter into its global and specific transcriptional regulation (Fig 2A). Previous approaches relied on normalization by synthetic constitutive promoters (Berthoumieux *et al*, 2013), prior parameterization of each promoter's global regulation by quantifying individual activities in the absence of specific regulation (Gerosa *et al*, 2013), or on a phenomenological scaling factor that captures pairwise differences

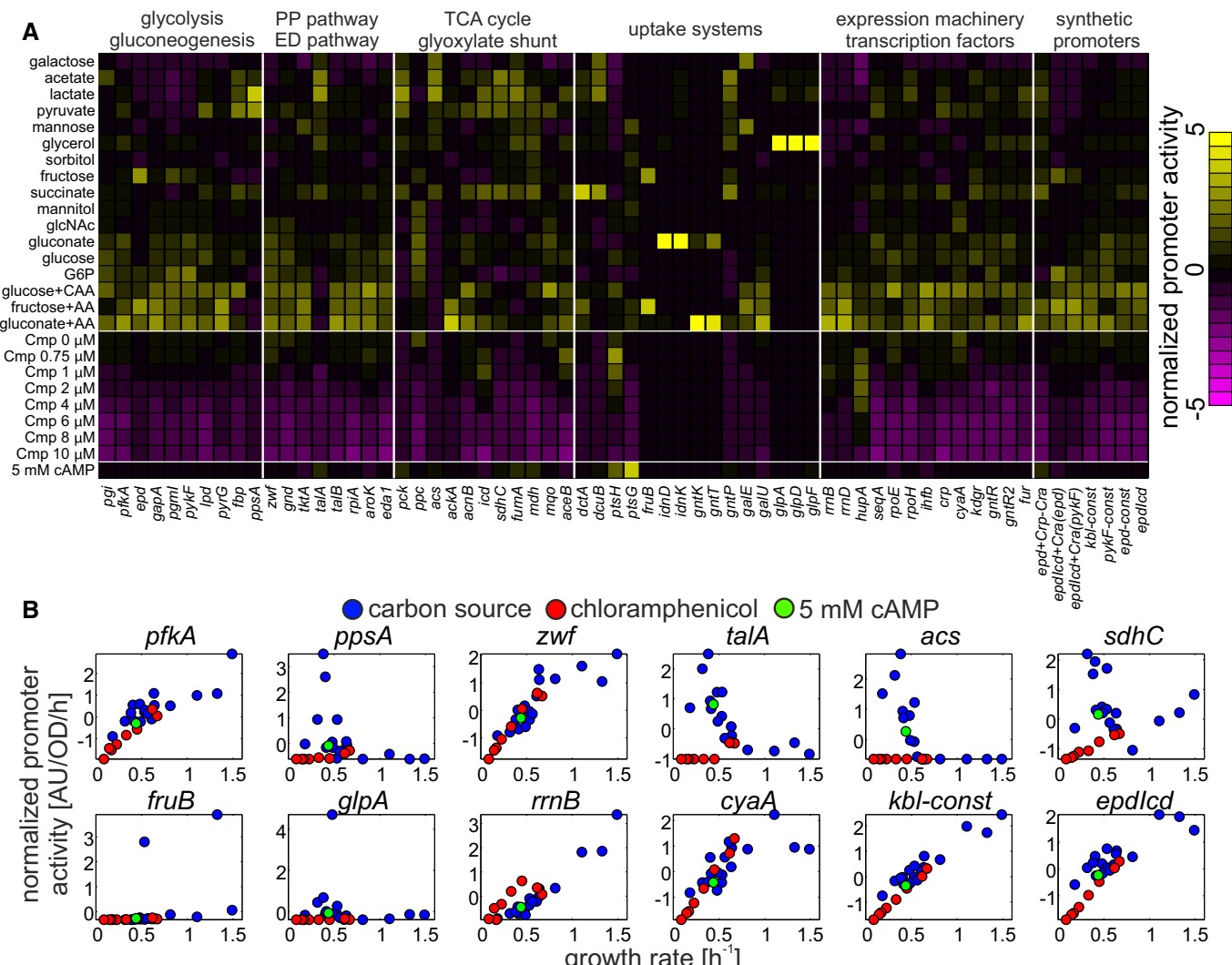

**Figure 1.** Steady-state promoter activity of 64 central metabolic genes across 26 conditions.

A   Steady-state promoter activities in various carbon sources and at different sub-lethal doses of chloramphenicol (Cmp), list of promoters given in Table EV1, and list of conditions given in Table EV2. Promoter activities were *z*-score-normalized to aid visualization, and promoters were grouped according to metabolic pathways and functional categories. Carbon sources were sorted by increasing growth rate (from top to bottom); chloramphenicol data were sorted by increasing chloramphenicol concentrations (from top to bottom). Last row: M9 glucose with 5 mM cyclic AMP (cAMP).

B   Activity of 12 selected promoters across all conditions plotted against steady-state growth rates.

in global regulation between conditions (Keren *et al*, 2013). To dissect global and specific transcriptional regulation across many conditions while retaining a mechanistic description of gene expression, we developed a mathematical description of promoter activity (pa; equation 1, see Appendix Text S1 for detailed description):

$$pa_{ij} = \left(E_j / K_{Ei}\right)^{\alpha_{Ei}} \cdot \prod_{l \in \{TF\}} \left(1 + TF_{lj} / K_{li}\right)^{\alpha_{li}} \quad (1)$$

Here, E denotes the activity of the expression machinery (in condition j) with its promoter-specific parameters $K_{Ei}$ and $\alpha_{Ei}$, and TF denotes the activity of each specific transcription factor (in condition j) that regulates the respective promoter with its promoter-specific parameters $K_{li}$ and $\alpha_{li}$. Upon transformation into

log space and normalization, for example, to a reference condition (marked with Δlog in the subsequent text), this equation can be approximated to (equation 2):

$$\Delta \log(pa_{ij}) = G + S$$
$$G = \alpha_{Ei} \cdot \Delta \log(E_j) \quad S \approx \sum_{l \in \{TF\}} \alpha_{li} \cdot \Delta \log(TF_{lj}) \quad (2)$$

As a result, log-normalized promoter activity can be described by a linear combination of global (formalized as expression machinery activity, denoted here as G) and specific (formalized as transcription factor activity, denoted here as S) transcriptional regulation. We hypothesized that global regulation would manifest as a common promoter activity pattern across conditions, similar to the

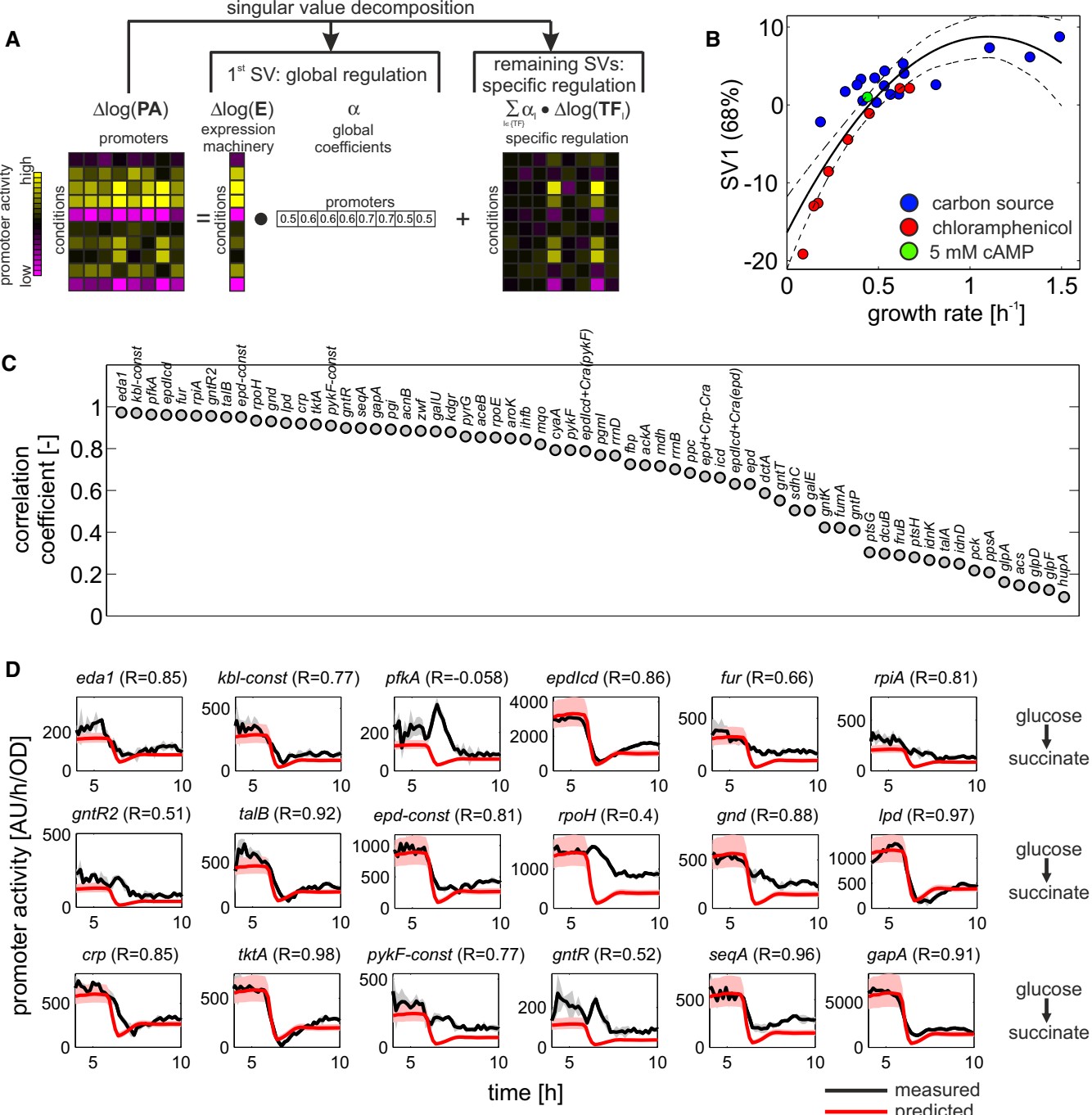

**Figure 2. Singular value decomposition of promoter activity data quantifies the contribution of global regulation.**

A  Schematic overview of the employed computational approach. Singular value decomposition was used to dissect quantified promoter activities into the contribution of global and specific transcriptional regulation. Here, global regulation is captured by the first singular vector, which accounts for the largest source of variance across conditions, whereas specific regulation is captured by the remaining variance, which cannot be explained by global regulation.

B  First singular vector (SV1, explaining 68% of the total variance) plotted against the respective growth rate. Black line: second-degree polynomial fit used for predictions in (D), dashed lines: 95% confidence interval of fit.

C  Pearson correlation between measured and reconstructed promoter activities across all promoters. Reconstruction was performed using only the first singular vector. Promoters were sorted according to their correlation coefficients. The majority of promoters are sufficiently explained by the first singular vector.

D  Comparison of predicted (red) and measured (black) promoter activity during a diauxic shift from glucose to succinate for the 18 promoters with the strongest input from global transcriptional regulation (see C). Data show the mean of three biological replicates, and gray shaded areas indicate the standard deviation across replicates for each time point. Dynamic SV1 values were calculated from growth rates using the polynomial fit in (B) and then reverted back to the original linear scale. Red shaded areas show the corresponding 95% confidence intervals. Pearson correlation coefficients between measured and predicted promoter activity time courses of each promoter are shown in brackets.

aforementioned scaling factor when comparing two conditions (Keren *et al*, 2013). Given the linearized description of promoter activity shown above, such patterns could then be inferred using linear decomposition approaches such as singular value decomposition (Alter *et al*, 2000) or principal component analysis (Bollenbach & Kishony, 2011). In this case, dominant global regulation would emerge as the first singular vector (i.e. the singular vector that captures most of the data set's variability), in analogy to the first principal component of the data set. Converse, specific transcriptional regulation would be the remaining variance that cannot be explained by the first singular vector (Fig 2A).

When applying singular value decomposition to the log-normalized promoter activity data, few common patterns, or singular vectors, captured most of the data set's variability (Appendix Fig S7). Importantly, the first singular vector showed strong growth rate dependence (Fig 2B, data in Table EV4), in line with our hypothesis that it captures the effect of dominant global regulation on promoter activity. Although the growth rate dependence of the first singular vector was particularly pronounced in the chloramphenicol conditions, which directly affect global regulation, we found that this dependence was also maintained when excluding these conditions from the analysis (Appendix Fig S8). To quantitatively assess how well the first singular vector explains the activity of individual promoters, we next compared measured promoter activities with their respective reconstruction purely based on the first singular vector. We found very good agreement particularly for the synthetic constitutive promoters (Fig 2C, data in Table EV4). These results suggested that the first singular vector indeed captures the growth-dependent impact of global transcriptional regulation. Conversely, we hypothesized that by measuring the growth rate in another condition and using the aforementioned relationship between growth rate and singular vector (Fig 2B), we should be able to predict the expression of constitutive-like promoters. We tested this hypothesis by measuring promoter activities during a diauxic shift from glucose to succinate (Gerosa *et al*, 2015). During this diauxic shift, cells transition from fast growth on glucose to slow growth on succinate, separated by a transient lag phase without growth. We compared measured promoter activities to predictions utilizing the relationship between measured growth rate and the first singular vector (Fig 2D, data in Table EV5) and found that promoters whose steady-state activity was dominated by the first singular vector (as determined in Fig 2C) were also well predicted during the diauxic shift. For some promoters, such as *pfkA*, *pykF*, and *rpoH*, measurement and prediction disagreed during the lag phase, presumably due to regulators that are only activated in response to the stress encountered during transient carbon starvation. One such regulator is the stress sigma factor RpoS, which is active during carbon starvation (Peterson *et al*, 2012) and has been reported to regulate these promoters (Salgado *et al*, 2013).

As a second validation experiment, we predicted the activity of central metabolic promoters in six additional conditions (measurements in Table EV3 and prediction in Table EV4) that do not constitute carbon source changes based on the respective steady-state growth rate and its relationship with the first singular vector (shown in Fig 2B). We found that not only constitutive-like promoters, but also in fact the vast majority of promoters could be predicted well (Appendix Fig S9). This result shows that our approach indeed allows to quantify growth rate-dependent global regulation, and suggests that global regulation has a strong impact on promoter activity also beyond carbon source changes.

Thus, the straightforward computational approach presented here enables dissection of global and specific transcriptional regulation based on singular value decomposition for large numbers of promoters without prior promoter-specific parameterization (Gerosa *et al*, 2013), and without the requirement for constitutive promoters to be used for normalization (Berthoumieux *et al*, 2013). The contribution of global regulation thus quantified is considerable, accounting for 68% of the total variability in promoter activities across conditions. Conversely, the remaining 32% of variability is the consequence of specific regulation.

## Systematic identification of metabolites affecting specific transcriptional regulation

Subtracting the confounding contribution of global regulation allows us now to investigate the specific, presumably transcription factor-driven regulation of each promoter. In particular, we were interested in potential regulatory metabolites and the transcription factors that ultimately exert the regulatory function on the promoters. To establish a link between promoters and potential regulatory metabolites, we started from the observation that in central metabolism, bacterial transcription factor activity is typically not regulated by changes in transcription factor expression itself (Ishihama *et al*, 2014; Gerosa *et al*, 2015) but rather post-translationally. For the frequent cases where direct binding of regulatory metabolites modulates transcription factor activity (Ulrich *et al*, 2005), a promoter's specific transcriptional regulation S can be described by the sum of metabolites M regulating its cognate transcription factors, weighted by two condition-independent and promoter-specific parameters $\alpha_{li}$ and $\beta_{lk}$ (denoting the impact of each metabolite on the respective transcription factor; equation 3):

$$S \approx \sum_{l \in \{TF\}, k \in \{M\}} \alpha_{li} \cdot \beta_{lk} \cdot \Delta \log(M_{kj}) \tag{3}$$

where i denotes the promoter and j denotes the condition. In principle, regulatory signals could therefore be identified by correlating the activity of a promoter's transcription factor activity to changes in potential regulatory metabolite concentration (Appendix Text S2). In practice, however, this inference is challenging because promoters are often regulated by several transcription factors and transcription factors may receive more than one regulatory signal. Since not all transcription factors/regulatory signals are likely to affect a promoter to the same extent in a given set of conditions, we first aimed for a systematic identification of promoters whose specific transcriptional regulation can be explained by single dominant regulatory signal. Given our focus on central metabolism, these regulatory signals are likely to be central metabolites. To test our hypothesis, we therefore quantified the concentration of 47 central metabolites during exponential growth in 23 of the 26 conditions by targeted metabolomics (Buescher *et al*, 2010; Appendix Fig S10, data in Table EV6). For each metabolite, we tested whether it can explain the specific transcriptional regulation component of any of the tested promoters based on the model described in equation 3 (Fig 3A). Specifically, we determined Pearson correlation coefficients between the measured specific

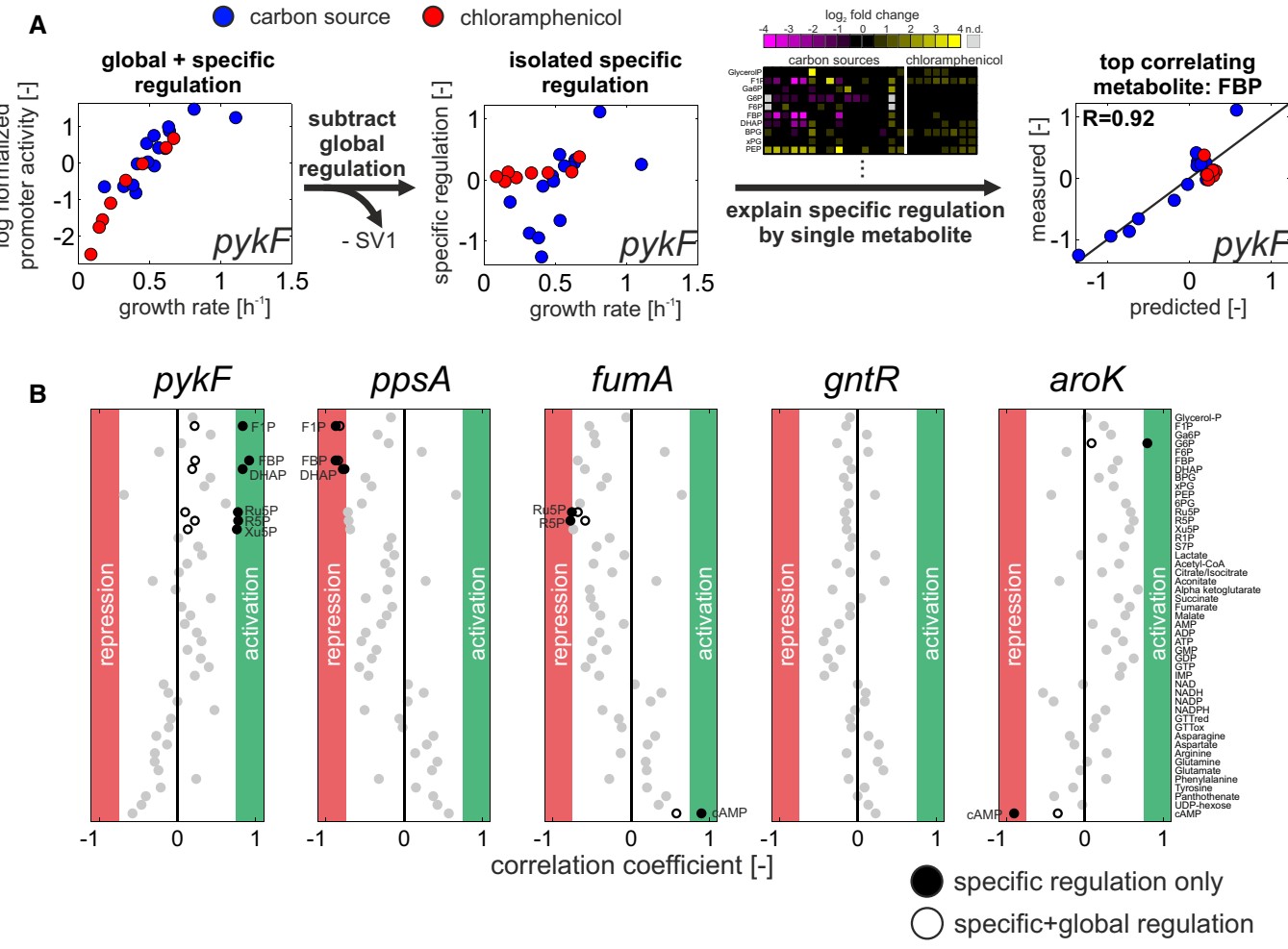

**Figure 3. Systematic identification of metabolites affecting specific transcriptional regulation.**

A Outline of approach using *pykF* as an example: First, global regulation is removed from each promoter by subtracting the first singular vector (SV1, see Fig 2A). Its remaining specific regulation is then related to each metabolite separately by linear regression to identify potential metabolic signals (goodness of fit assessed by Pearson correlation between measured data and the prediction based on its reconstruction).

B Summary of analysis for five promoters. Full circles: correlation coefficients when correlating each promoter's specific regulation with each metabolite across all conditions. Gray circles: metabolites that do not pass the correlation coefficient cutoffs [−0.75 < R < 0.75]. If metabolites pass the correlation coefficient cutoffs, their respective correlation coefficients without removal of global transcriptional regulation (combined output of global and specific regulation) are shown in empty circles.

transcriptional regulation and its reconstruction (obtained by linear regression based on equation 3, see Materials and Methods) to assess the explanatory power of each metabolite. Leave-one-condition-out cross-validation showed that in the vast majority of cases, the obtained correlation coefficients were not affected by omitting one of the tested conditions from the analysis (Appendix Fig S11). Similar results were obtained with a non-parametric method (rank correlation) to assess the agreement between measured and reconstructed data (Appendix Fig S12).

Typically, only one or few single metabolites could potentially explain each promoter's specific transcriptional regulation component (Fig 3B and Appendix Fig S13, full data in Table EV7). Cases with more than one potential regulatory metabolite were always based on strong cross-correlation of the metabolites across conditions. For example, the six metabolites identified for *pykF*, namely F1P, FBP, DHAP, Ru5P, R5P, and Xu5P, exhibited a median correlation of 0.83, making it difficult to differentiate which one(s) of them

are true signals. Conversely, most metabolites appear to be irrelevant for explaining specific transcriptional regulation of any of the promoters, and few metabolites were inferred to regulate more than one promoter. For example, F1P and FBP were identified as potential regulatory signals for several glycolytic promoters, and cyclic AMP was identified as a potential regulatory signal for a third of the tested promoters, in particular in TCA cycle and carbon uptake systems (Appendix Fig S13A).

In total, for about 50% of the promoters, at least one of the available metabolites could explain the specific transcription as a single input. When considering only promoters without a dominant contribution of global regulation (correlation coefficient $R < 0.75$ for reconstruction of the promoter based on global regulation alone, see Fig 2C), the percentage increased to 66%. To also systematically identify potential double metabolite inputs to promoter activities, we next tested all metabolite pairs for explaining the specific transcriptional regulation of each promoter. Statistical significance of

the improvement was assessed through the Akaike information criterion that accounts for different numbers of parameters in the model (Burnham *et al*, 2011; Link *et al*, 2013). In the vast majority of cases, no metabolite pair explained specific regulation better than the best single metabolite (Appendix Fig S14B). One of the few exceptions was the *cyaA* promoter, which was best explained by cyclic AMP and L-phenylalanine (Appendix Fig S14A). Thus, these results suggest that central metabolic promoters are mostly regulated by single dominating regulatory signals.

To obtain a quantitative, single-promoter resolution picture of the transcriptional program governing *E. coli* central metabolism, we next combined the metabolite-dependent specific transcriptional regulation with global regulation (Fig 4A, data in Table EV7). For the majority of promoters, the contribution of global regulation was large, suggesting that specific transcriptional regulation merely modulates a dominant global regulatory input. However, there were also cases with a strong contribution of specific transcriptional regulation, such as uptake systems and TCA cycle promoters that were mostly explained by cyclic AMP. Another exception was the gluconeogenic *ppsA* promoter, which was largely explained by FBP. Conversely, only the three metabolites cyclic AMP, F1P and FBP explained the majority of observed specific transcriptional regulation. Notably, only few promoters, such as those associated with gluconate utilization (*idnD, idnK gntK, gntT*), could not be explained by either global or specific transcriptional regulation, presumably because we did not quantify the underlying metabolic signal (see Appendix Fig S15 for individual reconstructions of each promoter based on the network depicted in Fig 4A). Leave-one-condition-out cross-validation further corroborated that these few metabolite signals were sufficient to quantitatively predict the activity of most promoters in the excluded condition (Fig 4C, Appendix Fig S16). Taken together, these results demonstrate that the transcriptional program of *E. coli* central metabolism is dominated by global regulation with few specific regulatory signals in response to nutrient variations or perturbations of the expression machinery itself.

Since we used GFP-based reporters, global regulation at the level of translation (Borkowski *et al*, 2016) could mask some of the transcriptional responses that occur only at the mRNA level. To explore this possibility, we compared published transcriptomics data (Gerosa *et al*, 2015) of central metabolic genes to our promoter activity data under eight common conditions. For 50% of the considered metabolic genes, expression changed little across conditions (Appendix Fig S17). Importantly, when considering those genes whose expression does change across conditions (at least one condition with a log2 fold-change < −1 or > 1), we found that their expression patterns agree well with the specific transcriptional regulation component of the corresponding promoters (Appendix Fig S18). This consistency suggests that our experimental data do capture most of the transcriptional regulatory response. Moreover, even in cases where the agreement between transcriptomics and promoter activity data was only moderate (e.g. *gapA, pykF, fumA* in Appendix Fig S18), we were still able to identify known regulatory metabolite signals (Fig 4A). One exception is malate synthase (*aceB*), for which transcriptomics and promoter activity data deviated substantially, possibly because some of the regulatory elements in its complex promoter (Salgado *et al*, 2013) may not have been included in the respective reporter plasmid.

## Relating metabolic regulatory signals to transcription factors

Finally, we asked which transcription factors mechanistically establish the inferred regulatory links between metabolites and promoters. Based on the known transcriptional regulatory network of *E. coli* central metabolism (Salgado *et al*, 2013) with over 30 different transcription factors (Table EV8), we determined the overlap between each metabolite's target promoters (as shown in Fig 4A) and the frequency at which these promoters were regulated by a given transcription factor (Fig 4B, data in Table EV7). Here, we focused on the four metabolites (F1P, FBP, PEP, and cyclic AMP) with more than two predicted promoter targets. Reassuringly, this analysis correctly predicted the well-known interactions between cyclic AMP and the transcription factor Crp (activation) and between the FBP and F1P and the transcription factor Cra (inhibition). In the case of PEP, no interacting transcription factor could be identified: The only transcription factor with an overlap in inferred promoter targets of PEP of over 50% was Crp, but this overlap was not statistically significant when accounting for the large number of reported Crp targets in central metabolism (Fig 4B). Given the correlation of PEP and the main Crp regulator cyclic AMP across conditions ($R = 0.55$), PEP likely constitutes a false-positive signal.

Notably, a considerable fraction of the promoters predicted to be regulated by cyclic AMP were either so far unknown Crp targets or responded opposite to the reported regulatory interaction. For example, our analysis predicted activation of *talA* and *pck* promoters by Crp-cAMP, in contrast to previous reports using reporter plasmids in Crp deletion (*talA*) or Crp overexpression strains (*pck*), respectively, which did not account for global regulation (Shimada *et al*, 2011; Nakano *et al*, 2014). We independently confirmed the *talA* and *pck* activation by Crp-cAMP through external supplementation of cyclic AMP, which increased promoter activity in the wild-type, but not in a Crp knockout strain (Appendix Fig S19A). Moreover, activation of these promoters in carbon sources supporting slow growth required Crp (Appendix Fig S19B). These results highlight the importance of taking global regulation into account when interpreting promoter activity measurements, especially when the growth rate is different between strains or conditions.

While our approach recovered many reported interactions (boxes with thick edges in Fig 4A), about 30 previously reported interactions were not recovered (gray boxes in Fig 4A), some of which had been validated *in vivo* (such as between *ppc* and Cra-FBP/Cra-F1P (Shimada *et al*, 2010), or between *pgi* and Crp-cAMP (Shimada *et al*, 2011)]. Since these *in vivo* validations were performed by GFP or LacZ reporter expression in transcription factor deletion mutants, it is conceivable that the observed expression differences were indirect consequences of altered global regulation in slower growing mutants. However, an alternative explanation could be that we missed these interactions in the process of removing the contribution of global regulation, for example, due to the singular value decomposition. To exclude this possibility, we focused on four previously reported metabolite–transcription factor pairs (GlpR-glycerol-P, Cra-F1P, Cra-FBP, Crp-cAMP) and examined the distribution of correlation coefficients across all reported target promoters with and without removal of global regulation (Fig 4D). Reassuringly, we did not detect any examples of reported promoter–metabolite interactions that were only recovered if the effect of global regulation on promoter activity was retained. Conversely, over 20%

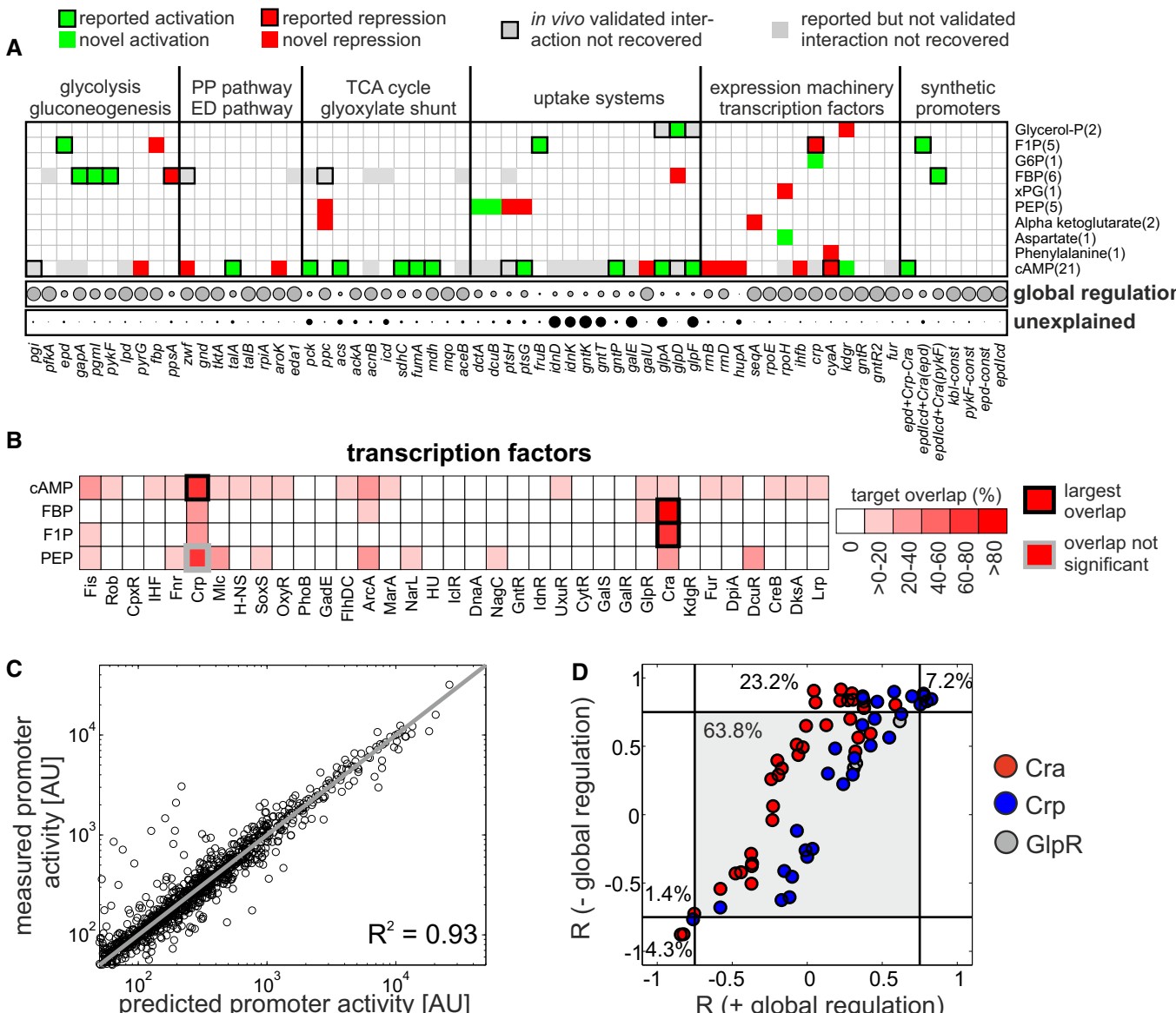

**Figure 4. Inferred global and specific transcriptional regulatory program of *Escherichia coli* central carbon metabolism.**

A   Inferred promoter–metabolite regulation network (Table EV7). Promoters were sorted according to metabolic pathways. Top panel: metabolite regulatory signals as determined in Fig 3 and Appendix Fig S14. Activating and inhibiting interactions are shown in green and red, respectively. Interactions that had been validated *in vivo* in previous studies [i.e. Shimada *et al* (2010, 2011)], but were not recovered in this work, are shown in gray with thick black edges. Only metabolites with at least one potential promoter target are shown. In parentheses: number of potential promoter targets for a given metabolite. Bottom panel: contribution of global transcriptional regulation across conditions for each promoter (gray), and unexplained part of the promoter response (black). Gray circle sizes denote the contribution of global transcriptional regulation to the respective promoter as determined in Fig 2B. Black circle sizes denote how much of the promoter's response remained unexplained after including metabolite signals (= 1 minus the correlation coefficient between measured promoter activity and reconstruction based on global regulation + inferred metabolite signals if applicable).

B   Inference of potential transcription factor–metabolite interactions based on the reported transcriptional regulatory network (as reported in RegulonDB; Salgado *et al*, 2013, Table EV8). For each metabolite with more than two target promoters, its target overlap with each of the transcription factors regulating central carbon metabolism was calculated (Table EV7). Thick black edge: transcription factor showing the largest target overlap with the respective metabolite. Hypergeometric testing (Zampar *et al*, 2013) was used to assess the enrichment of transcription factor targets among each metabolite's targets. All highlighted transcription factor–metabolite interactions showed significant enrichment (*P*-value < 0.1, Table EV7), with the exception of the interaction between Crp and PEP (*P*-value 0.14), which is shown with a thick gray edge.

C   Leave-one-condition-out cross-validation of the network in (A). $R^2$ denotes the overall goodness of fit between measured and predicted promoter activity across all conditions and promoters.

D   Correlation coefficients of reported promoter–metabolite interactions, which were obtained from RegulonDB based on four metabolite–transcription factor interactions (Glycerol-P–GlpR, F1P–Cra, FBP–Cra, and cAMP–Crp), without (−) or with (+) confounding global regulation. Percentage: fraction of reported interactions that falls into the respective sector (divided by black lines).

of reported interactions were only recovered after removing the confounding impact of global regulation (see also Appendix Fig S13A and B for a comparison of identified regulatory metabolites with or without confounding global regulation). This finding re-iterates that removing global regulation is pivotal for the identification of functionally relevant promoter–metabolite interactions. Notably, about 60% of reported promoter–metabolite interactions were not recovered in either case, suggesting that these interactions are not relevant under the tested conditions.

Overall, the results demonstrate that our approach systematically identifies *in vivo* relevant regulatory signals of transcriptional regulation and also the responsible transcription factors, provided the underlying regulatory network topology is at least partially known.

## Discussion

Here, we unravel the transcriptional program that governs *E. coli* central metabolism at an individual promoter level. Starting from measured activities of about 100 central metabolic promoters during steady-state growth under 26 environmental conditions, we identified global regulation by the growth rate-dependent cellular expression machinery as the dominant regulatory input for the majority of promoters, accounting for about 70% of the total expression variance across conditions. Specific, transcription factor-mediated regulation was confined to relatively few promoters, in particular in the TCA cycle. Interpreting the metabolome response under the same conditions with an approximate mathematical description of promoter activity, we further identified candidate metabolites that might serve as regulatory signals for the transcription factors. Our data-driven approach reveals a surprisingly simple transcriptional regulatory program of central carbon metabolism, in which global regulation, together with two transcription factors (Cra and Crp) governed by three regulatory metabolites (FBP, F1P, and cAMP), was sufficient to explain the majority of changes in promoter activity across conditions (Fig 4). Thus, this work provides a first quantitative map of the *in vivo* relevant mechanisms that are responsible for the coordination of central metabolic genes in *E. coli*.

The dominant role of global regulation is consistent with previous observations (Zaslaver *et al*, 2009; Berthoumieux *et al*, 2013; Gerosa *et al*, 2013; Keren *et al*, 2013) and may explain why transcriptional adaptation to environmental changes is typically accompanied by a large number of gene expression changes, even for very closely related conditions (Kao *et al*, 2004; Jozefczuk *et al*, 2010; Costenoble *et al*, 2011; Buescher *et al*, 2012; Nicolas *et al*, 2012). A limitation of our and many other GFP reporter-based studies is that the assessed global regulation becomes a conglomerate of transcriptional (Klumpp & Hwa, 2008) and translational effects (Borkowski *et al*, 2016). Recently, Borkowski *et al* showed that growth-dependent global regulation exerts its effect predominantly at the level of translation in *Bacillus subtilis* (Borkowski *et al*, 2016). Nevertheless, experimental evidence suggests that global regulation also affects transcriptomics data. For example, transcriptomics studies of yeast in various nutrient limitation experiments have shown that the expression of a large fraction of genes strongly depends on the cellular growth rate regardless of the exact type of limitation (Brauer *et al*, 2008).

Surprisingly, only two transcription factors, Cra and Crp, triggered by cyclic AMP, FBP, and F1P, suffice to explain a large

fraction of the specific transcriptional regulation. In particular, FBP/F1P inhibition of Cra regulates glycolysis and cyclic AMP activation of Crp regulates expression of TCA cycle and carbon source utilization pathways (Appendix Fig S20). These findings concur with previous studies highlighting the importance of Cra in regulating the switch between glycolysis and gluconeogenesis (Ramseier, 1996; Kotte *et al*, 2014) and for the sensing of glycolytic flux (Kotte *et al*, 2010; Kochanowski *et al*, 2013). Similarly, our results are consistent with the demonstrated importance of Crp for regulating TCA cycle fluxes (Nanchen *et al*, 2008; Haverkorn van Rijsewijk *et al*, 2011) and carbon utilization (Kaplan *et al*, 2008; Aidelberg *et al*, 2014). Notably, F1P and FBP affected different sets of promoters more strongly through the same transcription factor Cra. This difference in effector specificity appears to be largely encoded in the respective Cra binding sites, since addition of a Cra binding site from a F1P-regulated promoter rendered a synthetic promoter more specific for F1P, and vice versa (Fig 4A). Although structural information about Cra together with its effectors is limited, our results suggest that binding of F1P and FBP may trigger distinct conformational changes in Cra. Importantly, this work not only identifies Cra and Crp to be the two key transcriptional regulators of central metabolism using a data-driven approach, but also provides, for the first time, quantitative information on which genes they actually affect *in vivo*. Beyond quantifying the regulatory effect on known target genes, we found many instances of known regulatory interactions that do not seem to affect gene expression *in vivo* under the tested conditions and also identified potential novel interactions (Fig 4). These findings may aid future computational investigations of *E. coli* metabolism, which typically have to rely on reported regulatory interactions to assume a transcriptional network when constructing mathematical models (Kotte *et al*, 2010; O'Brien *et al*, 2013; Kremling *et al*, 2015; Jahan *et al*, 2016).

The effect of Cra on glycolytic promoters was rather weak, only modulating the dominant global transcriptional regulation. What could be the physiological relevance of such a modulating regulatory signal? One attractive hypothesis emerges when considering protein concentration (as the final output of gene expression; Fig 5). Proteins expressed from constitutive promoters exhibit a negative relationship between growth rate and protein concentration under varying carbon source availability (Klumpp *et al*, 2009). If such proteins are required at high concentration to enable high fluxes during fast growth, global regulation alone will lead to even higher concentrations at slow growth (Fig 5B, upper panel), placing additional burden on the cell. Since glycolytic carbon sources tend to support fast growth and result in higher FBP concentrations, regulation through Cra-FBP (i.e. repression by Cra which is alleviated by FBP) may counter this effect, causing more constant protein concentrations across different growth rates (Fig 5, middle panel). Conversely, the regulatory input of Crp-cAMP yields a previously described linear negative relationship between protein concentration and growth rate under carbon limitation for catabolic proteins (You *et al*, 2013; Hui *et al*, 2015; Fig 5, lower panel). Thus, few such regulatory signals may allow cells to coarsely allocate proteome resources based on the supported growth rate.

The simple transcriptional program identified here suggests that *E. coli* uses only a small fraction of its transcriptional regulation network in a given environment, which is consistent with recent observations (Zaslaver *et al*, 2009; Keren *et al*, 2013). How can this

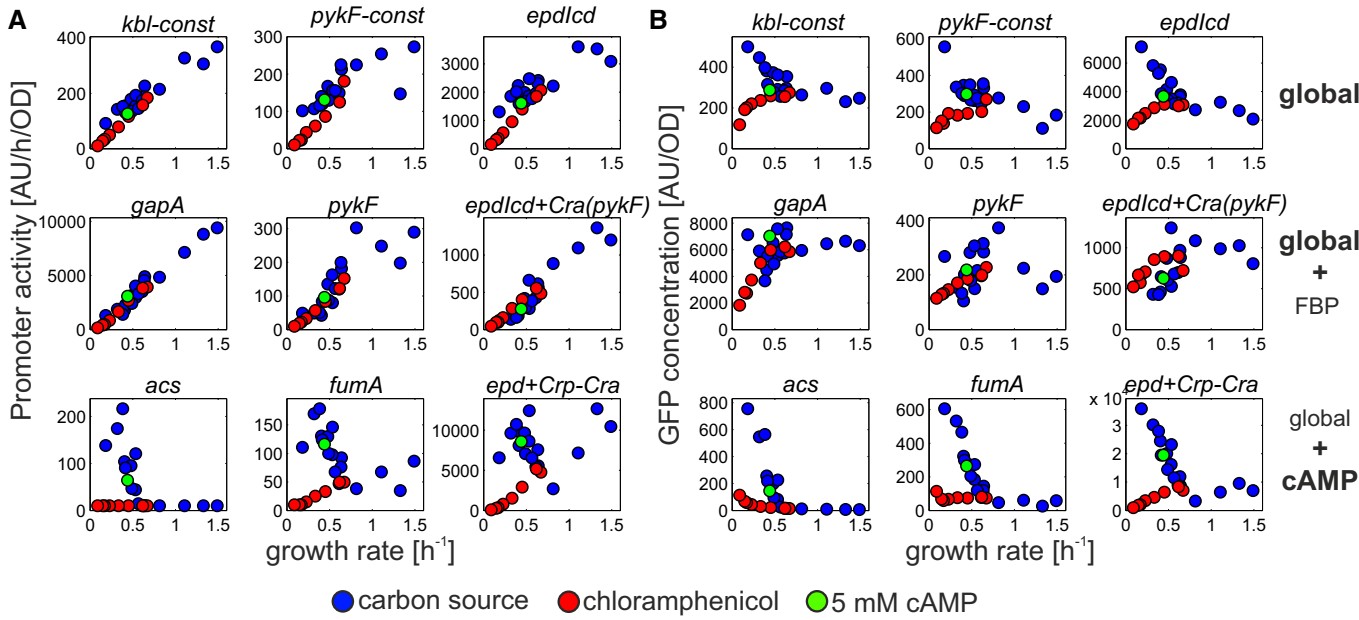

**Figure 5.  Relationship between promoter activity and protein concentration.**

A    Activities of promoters only affected by global transcriptional regulation (top panel), mostly by global transcriptional regulation with modulating specific input from fructose-1,6-bisphosphate (FBP, middle panel), and promoters with dominating specific transcriptional regulation through cyclic AMP (cAMP, lower panel).

B    Corresponding GFP concentration. GFP concentrations were calculated by dividing steady-state promoter activity by the respective growth rate.

finding be reconciled with the dense transcriptional regulation network of *E. coli* central metabolism that comprises over 30 additional transcription factors (Salgado *et al*, 2013)? Firstly, methods to physically map transcription networks, such as ChIP-chip (Cho *et al*, 2012), ChIP-seq (Furey, 2012), or SELEX (Shimada *et al*, 2010, 2011), typically provide binding information without assessing under which conditions binding occurs. Since our approach relies on detecting (metabolite-dependent) changes in transcription factor activity across conditions, we cannot capture changes that are not triggered by internal metabolite signals and changes that occur under other, not tested conditions. We certainly expect additional specific transcriptional regulation under different types of stress conditions, as already suggested by the few stress conditions tested here (Appendix Fig S9). Nevertheless, for the broad range of investigated carbon source conditions, we can explain about 90% of the observed transcription changes in central metabolism (Fig 4C, Appendix Fig S16) by the simple program, leaving only a relatively small unexplained portion to be subject to additional transcriptional regulation.

This proof-of-concept study focused on the relatively well-characterized central metabolism of *E. coli* (Chubukov *et al*, 2014). Our unbiased mathematical approach for systematic identification of potential metabolic regulatory signals can easily be extended to other cellular networks and organisms, in particular because it does not necessarily require information about the underlying regulatory network. Moreover, this approach only requires gene expression and metabolite data in matching conditions (which are becoming increasingly available) and is computationally straightforward. This work may serve as a template for data-driven systematic identification of novel metabolite regulatory signals, and ultimately transcription factor–metabolite interactions, from the correlation of steady-state data. The approach is not restricted to gene expression,

but can in principle also be used to infer metabolite regulators of protein kinases from metabolomics and phospho-proteomics data.

## Materials and Methods

### Reagents and strains

Unless stated otherwise, all reagents were obtained from Sigma-Aldrich. Fluorescent transcriptional reporter plasmids were directly obtained from Zaslaver *et al* (2006); Gerosa *et al* (2013) or constructed as described in the original study (Zaslaver *et al*, 2006), and subsequently transformed into the *E. coli* wild-type strain BW25113 (Baba *et al*, 2006). The *Crp* deletion strain was obtained from Baba *et al* (2006) and cured from its antibiotic resistance as described previously (Datsenko & Wanner, 2000). See Table EV1 for full list of promoters.

### Cultivation

All experiments were performed using M9 minimal medium (see Table EV2 for full list of conditions). Cultivations for the quantification of promoter activity were performed as described previously (Gerosa *et al*, 2013). Briefly, M9 medium batch cultures in 96-deep-well format plates (Kuehner AG, Birsfelden, Switzerland) were inoculated 1:50 from LB precultures and incubated overnight at 37°C under shaking. Subsequently, 96-well flat transparent plates (Nunc, Roskilde, Denmark) containing M9 medium (fill volume 200 ml) were inoculated 1:200 with overnight cultures and sealed with Parafilm to reduce evaporation. Online measurements of optical density at 600 nm ($OD_{600}$) and fluorescence (excitation

wavelength: 500 nm, emission wavelength: 530 nm) were performed at 37°C with shaking using a plate reader (TECAN infinite M200, Tecan Group Ltd, Männedorf, Switzerland) at 6- to 10-min (steady-state experiments) or 10-min (dynamic experiments) intervals. Diauxic shift experiments were performed as above, using M9 medium with 0.5 g/l glucose and 2 g/l succinate that was inoculated from M9 glucose precultures. Cultivations for the quantification of intracellular metabolite concentrations were performed as follows: M9 medium batch cultures in 96-deep-well format plates (Kuehner AG, Birsfeld, Switzerland) were inoculated 1:50 from LB precultures and incubated overnight at 37°C under shaking. Subsequently, 96-deep-well plate cultures were inoculated with overnight cultures to a starting $OD_{600}$ of 0.03 (total fill volume per well: 1.2 ml) and incubated at 37°C under shaking. Culture $OD_{600}$s were monitored by $OD_{600}$ sampling from parallel wells on the same deep-well plate and subsequent $OD_{600}$ measurements using a plate reader (TECAN infinite M200, Tecan Group Ltd, Männedorf, Switzerland).

**Quantification of intracellular metabolite concentrations**

Metabolomics samples were taken during mid-exponential phase at ODs between 0.5 and 0.7 by fast filtration (sampling volume: 1 ml; Link *et al*, 2013) and were immediately quenched in 4 ml quenching/extraction solution (40% methanol, 40% acetonitrile, 20% $H_2O$, all v/v) at −20°C (Link *et al*, 2012). To normalize for variations in sample processing, 100 μl of a fully $^{13}$C-labeled *E. coli* internal metabolome extract was added. Samples were incubated for 2 h at −20°C, subsequently dried completely at 120 μbar (Christ RVC 2-33 CD centrifuge and Christ Alpha 2-4 CD freeze dryer), and stored at −80°C until measurements. Before measurements, samples were resuspended in 100 μl water, centrifuged for 5 min (5,000 *g*, 4°C) to remove residual particles, and transferred to V-bottomed 96-well sample plates (Thermo Fisher Scientific). Measurement, data acquisition, and data analysis were performed as described previously (Buescher *et al*, 2010; Kochanowski *et al*, 2013). Briefly, separation of compounds was achieved by ion-pairing ultrahigh performance liquid chromatography (UPLC) using a Waters Acquity UPLC with a Waters Acquity T3 end-capped reverse phase column (dimensions, 150 mm × 2.1 mm × 1.8 μm; Waters Corporation) and coupled to compound detection using a tandem mass spectrometer (Thermo TSQ Quantum Ultra triple quadrupole; Thermo Fisher Scientific). Data acquisition and peak integration was performed with in-house software. To determine the absolute concentration of metabolites, a 1:3 dilution series of a standard solution (containing more than 80 metabolites of central carbon metabolism) with $^{13}$C internal standard was prepared and measured in parallel. Conditions in which a metabolite was secreted or used as a carbon source were omitted in the analysis.

**Data processing**

All data processing steps were performed with custom MATLAB software. Promoter activities and corresponding growth rates were determined as described previously (Gerosa *et al*, 2013). Briefly, raw GFP and $OD_{600}$ time courses from each well were corrected for blank GFP and $OD_{600}$ before cell addition and smoothed using a moving average window with size 3. From

these time courses, promoter activity and growth rate were quantified as $d$GFP/($d$t × OD) and $d$ln(OD)/$d$t by two-point finite difference numerical approximation. Promoter activity was corrected for fluorescence background by subtracting the corresponding signal of the promoter-less plasmid reporter strain p139 (Zaslaver *et al*, 2006). Promoter activity and growth rate values for steady-state growth were calculated as the average value in the time range visually identified as exponential phase. Error estimates of promoter activity measurements were performed based on day-to-day comparison as described before (Keren *et al*, 2013). Promoters whose activities were below the detection threshold (determined by the activity of a promoter-less strain) in all tested conditions were discarded for further analysis. One-dimensional hierarchical clustering of *z*-score-normalized (but not log-transformed) promoter activity data was performed using the Pearson correlation coefficient as the distance metric between promoters (cutoff: 0.225).

**Dissecting global and specific transcriptional regulation**

Promoter activity data were transformed using the natural logarithm and *z*-score-normalized, and singular value decomposition (Alter *et al*, 2000) was used (MATLAB function *svds*). Promoters were treated as variables, and conditions were treated as observations, yielding singular vectors that have the same dimension as the conditions. The first singular vector, which captures most of the data set's variability, was defined as global regulation. The specific transcriptional regulation component of each promoter was quantified by subtracting this first singular vector. Note that for the normalized data used here, singular value decomposition and principal component analysis (Bollenbach & Kishony, 2011) yield identical results: The first singular vector is equivalent to the first principal component.

**Identification of regulatory metabolites**

Metabolites for which absolute quantification was available were first transformed using the natural logarithm and then normalized by the mean concentration across all tested conditions. If no absolute quantification was available, metabolite concentrations were quantified relative to the M9 glucose condition and then transformed using the natural logarithm. Each promoter's specific transcriptional regulation component was related to each log-normalized metabolite by linear regression based on the equation s = p × metabolite, where s is the specific transcriptional regulation component and p denotes a parameter that is specific for each promoter and metabolite (corresponding to the lumped parameter ($\alpha_{li} × \beta_{lk}$) in equation 2) to be determined in the regression. The goodness of fit was determined as the Pearson correlation coefficient between specific transcriptional regulation component and the corresponding prediction based on the fitted parameter p and the metabolite concentrations using the MATLAB function *corr*. Conditions in which a metabolite was used as a carbon source, or secreted by the cells, were omitted. Identification of pairwise regulatory metabolites was performed by systematic linear regression of each promoter's specific transcriptional regulation component based on the equation s = p1 × metabolite1 + p2 × metabolite2, where p1 and p2 denote the promoter- and metabolite-specific parameters to

be determined in the regression. The goodness of fit was again determined as the Pearson correlation coefficient between measured and predicted specific transcriptional regulation. To assess whether any metabolite pair can explain the respective promoter's specific regulation component better than the best single metabolite, we calculated the difference in Akaike information criterion (AIC), which penalizes the number of parameters when comparing different models (Burnham *et al*, 2011; $\Delta \text{AIC} = \text{AIC}_{\text{bestSingle}} - \text{AIC}_{\text{metabolitePair}}$). The AIC of each promoter–metabolite (or metabolite pair) combination was calculated as described before (Link *et al*, 2013):

$$\text{AIC} = N \cdot \log\left(\frac{\text{RSS}}{N}\right) + 2K \qquad (4)$$

Where $N$ denotes the number of conditions, RSS denotes the sum of squared residuals (between measured and predicted data), and $K$ denotes the number of parameters in the respective model ($K = 1$ for single metabolite, $K = 2$ for metabolite pair). See Appendix Text S3 for a more detailed description of the used algorithm to identify regulatory metabolites.

### Inference of transcription factor–metabolite interactions

Inference of potential transcription factor–metabolite interactions was performed based on the reported transcriptional regulatory network (as reported in RegulonDB; Salgado *et al*, 2013). First, for each metabolite that was identified as a potential regulator of more than two promoters, its target overlap with each of the transcription factors regulating central carbon metabolism was calculated (100% means that all of the target promoters of a metabolite are regulated by the respective transcription factor). Based on these calculations, the transcription factor with the largest overlap was selected. For these selected transcription factor–metabolite pairs, the enrichment of transcription factor targets among each metabolite's targets was assessed by hypergeometric testing (Zampar *et al*, 2013) to account for differences in the number of target promoter between transcription factors. Target enrichment was only calculated for the transcription factor with the largest target overlap, since it is error-prone for small numbers (i.e. 1–2) of target promoters: In these cases, enrichment analysis will heavily favor transcription factors with few (or only one) targets.

### Leave-one-condition-out cross-validation

The parameters of each identified promoter–metabolite interaction (see Fig 4) were re-fitted while omitting the data point belonging to the excluded condition. Using these re-fitted parameters, each promoter's specific regulation component in the excluded conditions was then predicted based on the respective metabolite concentration. Finally, each promoter's summed contribution of global and specific regulation was reverted back to linear scale. This procedure was repeated for each condition.

### Data availability

The data generated in this work are available as Tables EV1–EV8.

**Expanded View** for this article is available online.

## Acknowledgements

We thank Elad Noor for helpful discussions. We thank Fred Allain for access to the D-BIOL Biomolecular NMR Spectroscopy Platform at ETH Zurich.

## Author contributions

KK, LG, and US designed the study. KK and LG performed experiments and computational analyses. SFB and DC assisted with experiments. YVN performed additional experiments not included in the manuscript. KK and US wrote the manuscript, with contribution from LG.

## Conflict of interest

The authors declare that they have no conflict of interest.

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
