## [Review Process File · Molecular Systems Biology]

Few regulatory metabolites coordinate expression of central metabolic genes in *Escherichia coli*

Karl Kochanowski, Luca Gerosa, Simon F Brunner, Dimitris Christodoulou, Yaroslav V. Nikolaev, Uwe Sauer

Corresponding author: Uwe Sauer, ETH Zurich

Review timeline:

First Submission:	20 June 2016
Editorial Decision:	04 August 2016
Second submission:	19 October 2016
Editorial Decision:	14 November 2016
Revision received:	27 November 2016
Accepted:	30 November 2016

Editor: Maria Polychronidou

Transaction Report:

1st Editorial Decision

04 August 2016

Thank you again for submitting your work to Molecular Systems Biology. We have now heard back from the three referees who agreed to evaluate your manuscript. As you will see below, the referees raise substantial concerns on your work, which, I am afraid to say, preclude its publication in Molecular Systems Biology.

The referees appreciate the extensive data generated in the study. While reviewer #3 is cautiously positive, reviewers #1 and #2 point out that the conceptual novelty remains limited and raise significant concerns related to the data analysis and interpretation. Both reviewers indicated that they do not support publication of this work in Molecular Systems Biology

We have also circulated the reports to all reviewers as part of our 'pre-decision cross-commenting' policy.

During this process, reviewer #1, mentioned:

"Generally, I agree with the other reviewers' comments. Reviewer 3 noticed an interesting point that had escaped my attention on the first reading: most of the correlation of the first SVD vector with growth rate is due to the chloramphenicol data. This antibiotic inhibits ribosomes. I am therefore even more worried about the possibility that what we see is not transcriptional regulation, but modified translation because of ribosome activity. In other words, the dominance of global transcriptional regulation could be an artifact. In this respect, the experiments suggested by reviewer 2, using chemostat cultures instead of diauxic shifts in batch culture, would probably be rather informative by decoupling growth rate from carbon source or ribosome activity. Such experiments would also add much more novelty to the work, but would require a very considerable amount of

work; this would become an altogether different manuscript."

Moreover, reviewer #2 mentioned: "In addition to the concerns regarding novelty, reviewer #1 shares my concern regarding use of GFP for this particular study. Reviewer #1 also brings up another important concern that all constructs are likely to be not transcriptional fusions -that is a major issue if true. These three concerns together make me uncomfortable recommending this manuscript for publication."

Overall, considering the rather substantial concerns raised by the reviewers in combination with our journal policy that allows in principle a single round of major revision and the fact that the outcome of the additional experimentation suggested by the reviewers is unclear, we see no other choice but to reject the manuscript at this stage.

Nevertheless, considering that the reviewers did have positive words for the goals and potential value of the data generated in the study, we would be willing to consider a new and extended manuscript based on this work, provided that additional experimental analyses along the lines of the reviewers' suggestions are included and that the issues raised are convincingly addressed. We recognize that (as Reviewers #1 and #2 point out) this would involve substantial additional experimentation with unclear outcome and, as you probably understand, we can give no guarantee about its eventual acceptability.

If you do decide to follow this course then it would be helpful to enclose with your re-submission an account of how the work has been altered in response to the points raised in the present review.

I am sorry that the review of your work did not result in a more favorable outcome on this occasion, but I hope that you will not be discouraged from sending your work to Molecular Systems Biology in the future. In any case, thank you for the opportunity to examine this work.

REFEREE REPORTS

Reviewer #1:

The manuscript by Kochanowski et al. describes measurements of the promoter activity of almost 100 genes of *E. coli* using fluorescence reporter constructs in 26 environmental conditions that change the growth rate of the bacterial culture. These promoter activities are interpreted as a combination of global regulation (mostly transcriptional regulation) and specific regulation. The quantitative measurement of metabolites of central metabolism allows identification of the metabolic signal that triggers the specific regulation. The results confirm and extend the results of previous publication by the same authors and other laboratories: most of the transcriptional response of *E. coli* to changes in the environment is due to global effects and strongly correlated with growth rate. In addition, the authors show in this manuscript that very few metabolic signals, essentially two, are sufficient to explain more than ninety per cent of the transcriptional response of genes involved in central metabolism to environmental changes.

Globally, the experiments are well carried out and the analyses of the results are well done. In particular, the simplifications introduced in the mathematical analysis and the use of singular value decomposition for distinguishing between the major contributions to regulation (in this case, specific versus non-specific, growth-related effects) are adequate for the goal and message of the manuscript.

Some technical issues could be addressed more closely or commented on in the text.

1. The authors insist that they measure transcriptional regulation. However, since they use fluorescent reporter genes to assess promoter activity, some global effects could also be mediated by changes in translation efficiency.
2. The reporter plasmids used are taken from the promoter collection of Zaslavler et al. (2006). These constructions are not strictly speaking transcriptional fusions since they contain part of coding sequence of the gene that is replaced by *gfp*. Again, this could lead to regulatory effects other than purely transcription. Furthermore, a number of reporter plasmids have been constructed in this study. The sequences of these constructs should be given in the SI.
3. The key parameter of the study is growth rate which was estimated from a "time-range visually identified as exponential phase". A simple figure in the supplementary material should show one

example to illustrate the procedure.

4. The observation that parameters obtained from steady-state measurements can predict dynamical transitions (Figure 2D) should be commented. This result implies that changes in global transcriptional regulation are fact compared to the time-scale of a diauxic shift.

In summary

The results presented here are a follow-up of previous work. At least three previous publications in MSB (Berthoumieux et al., 2013, Gerosa et al., 2013, and Keren et al., 2013), all cited in the present manuscript, have demonstrated that global regulation dominates over specific regulation. The first reference used detailed dynamical analysis on a limited set of promoters; the second, by the group submitting the present manuscript, shows growth-rate dependence of global regulation in steady state as well as dynamical transitions on a larger set of genes; the third reference shows a similar result on a genome wide-level at steady state in different growth conditions. The present work thus confirms these previous publications for other promoters and slightly different experimental conditions.

The measurement of metabolites in the different growth conditions adds interesting information about the metabolic signals that are responsible for specific regulation of genes involved in central metabolism. The major conclusion of these measurements is the observation that very few metabolites, essentially cAMP and fructose-bis-phosphate (FBP) account for most of the observed, specific regulation. The dominant effect of cAMP and FBP (and two other metabolites) had been established in 2010 by a modeling approach, also published in MSB (Kotte et al., 2010).

The work presented in this manuscript thus confirms the conclusions of four previous publications in MSB and does therefore not add much novel insights or conceptual advances. In other words, the manuscript lacks originality and novelty. However, the information presented is still useful and the experiments, as well as the analysis of the results, are well carried out.

Reviewer #2:

Summary. Kochanowski et al. present detailed analysis of regulation of ~100 genes of central metabolism in *E. coli* by 30 TFs in 26 environmental conditions. They constructed a library of promoter-GFP constructs for each gene and investigated the activity of the promoters by assaying GFP fluorescence during growth in media that differ in nutritional composition (primarily C-source). They go on to explain differential expression of genes as a function of global and specific regulation. With respect to the latter, they conduct metabolite profiling to elucidate which metabolites are responsible for the condition-specific regulation of promoters whose activity could not be explained entirely by global regulation, which their data suggest is directly correlated to growth-rate and attributable to regulation by just two transcription factors (CRP and Cra). The metabolite profiling implicated cAMP, fructose-1-phosphate and fructose-1,6-biphosphate as major influencers of transcriptional regulation of central metabolism genes. Based on the results from this study they claim that few inputs regulate the transcriptional response of central metabolism in *E. coli*.

Review. Generally speaking, the manuscript is well written and easy to read. There were some glaring omissions in the introduction and discussion, which need to be addressed. For instance, work on systems biology of transcriptional regulation in *E. coli* is largely ignored -they should fix it to give a more unbiased view of prior work. But otherwise, the intro was well written and it introduces the necessary concepts to understand the presented results. The Methods section is well-written and detailed. Translation of different biological influences into a mathematical formalism was done elegantly. I also enjoyed reading the precise and robust justification for the choice of different thresholds used in the study (section 2 of Sup. Info.). Additionally, some negative results are also communicated which is really appreciated. However, I did not find the general approach of the work to be radically original or novel. I also have major concerns about the conclusions -especially, I find the evidence to be insufficient to support the claims. Even if the claims were supported, I am not particularly convinced that the insights are novel -in the limited context in which they might be relevant (i.e., diauxic shift).

The authors make a big claim that a significant proportion of transcriptional regulation of most central metabolism promoters is explained by growth rate -a claim that needs to be supported by compelling evidence. For reasons described below, more experiments are needed to support that

claim. A better experiment design would be to take away the effect of growth rate by growing cultures in chemostat mode, and then evaluate the consequence of a shift in C-source. This is admittedly a complex experiment design that is not easily scalable to many promoters -but I would like to see it done for at least few promoter-reporter construct strains. In particular, I would like to see how temporal changes in transcript levels of some of the 100 selected promoters correlate to changes in GFP fluorescence during a shift in C-source. The chemostat experiment design will help to address this major question that is central to all of the conclusions and claims: does turnover dynamics of GFP capture subtle changes at the level of transcriptional regulation? This is especially a concern for genes that experience transient up or down regulation at the transcript level that is unlikely to appear at the GFP level. Hence, it is not surprising that majority of regulation is correlated to growth rate -since protein synthesis of GFP will depend on abundance and availability of ribosomes. By contrast, subtle changes at the level of transcriptional regulation will be somewhat masked by the dominant influence of growth rate-correlated protein synthesis.

It is also not surprising that cAMP, Fructose-1,6-biphosphate, and fructose-1-phospate explain gene regulation of most genes of C-metabolism. Whereas consumption of the two phosphorylated sugar derivatives would be expected to reflect C-uptake rates through glycolysis, the level of cAMP is a reflection of the overall energy status (ATP/ADP ratio) of the cell -both of which are directly proportional to growth rate. These issues worry me that the observations are confounded by the flawed experiment design that expectedly would have catabolite repression and growth rate dynamics as dominant factors that mask all other transcriptional inputs. The claims are bound to be true only in the specific context of diauxic shift from one C-source to another -in which case, the results are not surprising and something that has been known for quite some time.

Another point that was surprisingly not discussed was the generally accepted view that regulation is not just about turning up or down genes independent of each other, rather it has to do with coordinating sets of genes that carry out related functions in relevant environmental conditions. Their claim is extraordinary in that "...a surprisingly simple regulatory program that relies on global transcriptional regulation and input from few intracellular metabolites appears to be sufficient to coordinate *E. coli* central metabolism, and explain about 90% of the experimentally observed transcription changes in 100 genes." I am especially curious to know why there are 30 transcription factors maintained in the *E. coli* genome - all apparently implicated in regulation of central metabolism genes -if only two transcription factors are sufficient. The authors hand-wave that these transcription factor might function in other stress conditions. This statement could be investigated further by mining the extensive compendia of publicly available transcriptomes for *E. coli*. In that regard, it was puzzling why other bodies of work that have looked at conditional operation of the *E. coli* gene regulatory network from a systems biology perspective were notably omitted from the introduction and discussion.

Other specific issues:

1. The number of conditions they list (26) is a bit overstated, as this number also includes control conditions in which cultures were grown with chloramphenicol (based on my count the correct number should be 17 different growth conditions).
2. At several places, (L43, L48, L317), authors use the term "adaptation" when I assume they meant "acclimation". Authors refer to the process by which cells adjust to a gradual change in its environment not to the dynamic evolutionary process by which a trait evolves in species populations. Authors should address this confusion.
3. Figures 1, 2b and 5 specifically highlight the condition "5 mM cAMP". Why? What is the rationale? Further background about the importance of that condition is expected. Please elaborate.
4. Paragraph L343-357 is generally difficult to understand. Please rewrite for clarity. Specify the major point and provide with sufficient background such that a broader audience will understand the message conveyed and the relevance of the finding.
5. L378-L380. Last sentence is irrelevant. Please remove. A conclusion tone sentence which summarizes the work and its relevance is advised.
6. In the introduction, I would recommend to discuss the role of promoters in the regulation of metabolism.

7. L103-L105. I recommend to improve the readability of the sentence. How many reporter plasmids were used initially? At first glance, the relationship between the genes and the promoters was not clear.
8. L126-L128. Hard to understand. What is the message of the sentence? Please rephrase.
9. L261. If PEP is a false positive, are all others with lower target overlap also false positives? Why? Please discuss or elaborate.
10. L311-314. It's not clear to which 2 TFs and 3 metabolites authors refer to. Please name them for clarity.
11. Syntax used in L449 ("promoter- and") and L456 ("promoter- and") is unclear. Please rewrite sentences for clarity.
12. L486. Based on what criteria are those 12 promoters selected. Please explain.
13. Figure 4. Some of the labels touch the grid lines, please fix.
14. Figure 1A. I suggest to order columns in descending order (from gluconate+AA to galactose) to maintain consistency. I leave this suggestion to the choice of authors.

Typos

1. L31. A comma is missing between "Remarkably" and "cyclic".
2. Eq. 2. Second term, S. At the end of equation, subindex of TF should be lj, not li.
3. L268 of Supp. Info. There should be a space between value and units in "2g/L".

Reviewer #3:

This study investigates the regulation of expression, both global and specific in central carbon metabolism. It shows the dominant role of the global response in central carbon metabolism. One of the impressive features of the study is the ability to systematically find metabolites that regulate transcription factors activity. I found the work timely and useful. My suggestions for improvement are detailed below.

Major points:

L. 152-154: This is where I think most readers will lose the ability to follow what was done. "When applying singular value decomposition on 153 the log normalized promoter activity data, few common patterns, or singular vectors, captured most 154 of the data set's variability (supplementary figure 4)."

SVD is not a simple thing and an intuitive explanation will be useful. It took this referee a long time to understand the axis, and so probably also to other people who will aim to really follow the details of what you did. What are then the relevant "axis" of the PCA/SVD? You should take the reader by the hand on this very challenging turf.

The authors find that the first and quite dominant SVD axis is strongly related to growth rate. They state: "As hypothesized, the first singular vector showed strong growth rate dependence (figure 2B)."

I think this is a key point and several things should be done to clarify it to the reader.

A. take the reader by the hand in explaining what are the steps done to achieve this result. Is it a correlation of the values in the 1st SVD direction of each condition point to its growth rate? This requires explanation at least to this reviewer.

B. What is the shape of the "dependence"? Can you plot it? You mention also doing a z

transformation to the log PA but is the dependence with or without the z transform?

C. The slope of the correlation before the z transform seems to me to be quite informative (unless I am misunderstanding, which frankly could be the case), if the slope of the 1st SVD axis to the growth rate is close to 2 it might suggest an underlying model. If it is quite different than 2 that is also quite interesting to point out somewhere. I might be off here.

This study has a wealth of data that I would imagine some researchers would like to analyze further with their own hypothesis. It will good if the authors publicly provide the raw data measured as well as the processed data and the code for the data analysis.

Figure 1/S3: The most clear distinction that I saw was between the chloramphenicol treatment conditions and all the rest. I feel this not so common condition swamped all the other things and was too bad given my interest in the "normal" growth conditions. I would urge the authors to also make a plot without the chl treatments that will appear in the SI if not the text. If indeed this has an effect as seems to me by eye, this issue should be pointed out and discussed in the text explicitly.

Figure 2A: Much of the explanatory power of the SV1 seems to come from the chloramphenicol conditions according to my understanding of the figure. What would be the results without these conditions? The authors should analyze and at least reflect and mention explicitly this point.

Around L. 200: Can you give a value of how much of the variability in expression that is specific you could explain with the top 1, 2, 3, 4 metabolites modulators?

330-332: "In particular FBP/F1P inhibition of Cra regulates glycolysis and cyclic AMP activation of Crp regulates expression of TCA cycle and carbon source utilization pathways." This nice view would gain a lot in my eyes from having a figure associated with it that shows the model of regulation of central carbon metabolism proposed by the authors in one simple schematic as the discussion suggests. I think this can become textbook material if done properly, and I would surely find it useful personally for me and my students.

It is not clear to me if the global response should be thought of as "global regulation" or as a passive response that is an outcome of resource allocation and thus is not an active act of "regulation". My personal view is the latter but in any case I think the reader can gain from at least mentioning this distinction in the discussion.

Minor points:

In lines 138-140 where you introduce the functional form of eq. 1 the reader can be perplexed about why there is the alpha exponents. It will be useful to explain that this is not for mechanistic reasons (as far as I understand) but mostly driven by mathematical convenience for the next step where everything will be analyzed in log space and thus this will lead to a solvable linear relationship.

144, eq.2, the $\Delta(\log(p_{ij}))$ is not defined so clearly. I would write as $\Delta(\log(p_{ij})) = \log(p_{ij}) - \log(p_{i0}) = G + S$

Also in Eq.2 the \sim relation for S is far from being immediately clear. Need to explain.

SI line 22, "TFconc l_j " I would consider denoting this as $c_{TF_{i,j}}$

L 27-28 "approximating each term expressed as $\log(1+x)$ with $\log(x)$ " should write when is this reasonable ?

L 31-32 "assuming that transcription factor expression does not change significantly across conditions". This is a very strong assumption. I do not know how well it holds but in any case the fact that you are using it should be made more explicit in text rather than just in the SI.

Many readers have heard of PCA but not SVD. It will be good to more clearly state their intimate relationship (synonyms for the purpose of this paper?).

Equation 2: shouldn't S, G also have subscripts of i/j ?

Equation 3: alpha is defined above equation with one index and in equation with two. It will also be good to explain again what i and j are in this context.

217-218: "In total, for about 50% of the promoters at least one of the available metabolites could explain the specific transcription as a single input." How do the authors define "could explain"? is there a numerical definition?

232-233: "and scored the improvement in agreement between measured and predicted promoter activity over global transcriptional regulation alone (figure 4A)." I do not think this is shown in 4A.

270-272: "These results highlight the importance of taking global regulation into account when interpreting promoter activity measurements, especially when the growth rate is different between strains or conditions." I liked this observation which I think is very insightful. The authors may want to mention a recent approach (PMID:27073913) that highlighted the utility of using the growth rate dependent global response as a null model on which to detect specific responses.

275-276: "(such as between ppc and Cra (Shimada et al, 2010), or between pgi and Crp (Shimada et al, 2011))", From the text I thought you are talking about promoter-metabolite interactions but the examples are between TF and promoter it seems. Please clarify.

284-286: "Reassuringly, we did not detect any examples of reported promoter metabolite interactions that were only recovered if the respective promoter's global regulation component had not been removed." There are double or triple negative here that was hard for me to parse.

313-314: "was sufficient to explain the majority of changes in promoter activity across conditions." How much is the majority? Will be useful to have a number

331 "Cra regulates glycolysis and cyclic AMP activation of Crp regulates expression of TCA cycle and carbon source utilization pathways." Can the word "regulates" be changed to something more concrete like up-regulates or down-regulates?

346-348: "For promoters that are solely subject to global transcriptional regulation, variation of carbon source availability yields a negative relationship between growth rate and protein concentration." I see this in the figure (5B??) but what is the reason for this? Was this predicted from your model? Even if not, that is worth stating explicitly for the benefit of the readers so they know what is understood and what is not understood at this time.

369: "for the broad range of here investigated" typo

435-440: It will be useful to say how many promoters were tested and how many could be reliably quantified above the background on average across the different conditions.

441-445: For the SVD, what are the dimensions of the space? Help the reader by explaining explicitly if the axis are promoter activities of each gene and the points are the conditions, or the axis are the promoter activities in each condition and the points are promoters. It is all trivial to the authors I imagine but for the readers it can be challenging.

Figures 1-5: the red/blue/green data points often overlap very much. I think that a bit smaller points would help the presentation. Removing the outer black lining around them would probably also help discern the colors.

Figure 2: The color red means different things in panels B, C, D. You might want to update this. A similar things happens in Figures 3 and 4.

I did not find an overall description of the variability in promoter activity, what is the average CV of a promoter across conditions? does it depend on the expression level?

Comparison between the promoter activities and other data sets, where data already exists (e.g. Schmidt+Heinemann proteomics) can be of interest.

If I understood correctly, the global effect is opposite to that predicted by Klumpp 2009. This is worth pointing out explicitly.

It is interesting that 70% of the variability is explained by growth rate where there is usually 10-20% of noise in such measurements. Might be good to mention something about noise level.

Figure S5 caption: state reference to full data. Mention the basis of the log is 10.

Second submission

19 October 2016

Main concerns raised by reviewers:

- 1. A major concern is that our GFP-based transcriptional reporter library is also affected by global regulation at the level of translation (see Borkowski et al., 2016. MSB 12: 870) and thus may not capture all of the regulation at the level of transcription.**

To address this concern, we performed the following additional analysis: first, we analyzed transcriptomics data from central metabolic genes under 8 different carbon source conditions from a recently published data set (Gerosa et al, 2015. Cell Systems). As expected based on our promoter activity data, the expression of the majority of genes changed less than two-fold across conditions (supplementary figure 16), suggesting indeed a moderate impact of specific transcriptional regulation in central metabolism. Second, we compared these transcriptomics data to our promoter activity data under the same conditions to assess whether we capture those expression changes that do occur. We found that after applying our computational approach to remove the effect of global regulation, the remaining specific regulation recapitulates the transcript abundance changes of most tested genes very well (for 62% of the tested genes, the correlation between transcript abundance specific regulation has a p-value < 0.05, see supplementary figure 17 for individual plots), with few exceptions that we discuss in the revised manuscript. Moreover, even in cases where the agreement between transcript and promoter activity data is only moderate (e.g. *fumA*, *pykF*, *gapA*), we are still able to identify known regulatory metabolite signals, showing the robustness of our approach. We included this analysis in the revised manuscript (main text L267-281). We also made clear in the text that in our experimental data global regulation is the cumulative result of transcription and translation (L350-357), and changed the term “global transcriptional regulation” to “global regulation” throughout the text.

- 2. A related concern is that the dominance of global regulation in our data may be mostly due to the selected conditions (i.e. chloramphenicol treatment, which directly affects translation and therefore global regulation).**

To alleviate the reviewers’ concern, we quantified the contribution of global regulation while excluding the chloramphenicol conditions. Although this contribution is reduced, it still accounts for over 50% of the total variance in the data set, suggesting that the strong impact of global regulation is not an artefact of the tested conditions. We included this analysis in the manuscript (supplementary figure 7).

- 3. The reviewers requested additional experiments to strengthen our claim regarding the dominance of growth rate-dependent global regulation, in particular by quantifying promoter activity while maintaining a constant growth rate (i.e. in chemostat cultures).**

As the reviewers acknowledged themselves, using chemostats to compare promoter activities in different conditions while keeping the growth rate constant is technically quite challenging and laborious given the large number of promoters considered here. Moreover, the added value of such an experiment is not clear to us. First, our present experimental data set already includes pairs of conditions with highly similar growth rates. The fact that in these condition pairs the vast majority of promoters have very similar activities re-iterates that (growth-dependent) global regulation indeed has a strong impact on promoter activity (see additional figure at the end of this response letter). Second, a major achievement in our study is the development of a scalable method to quantify global regulation across a wide range of different growth rates. In the initial submission of the manuscript, the prediction of promoter activity of constitutive-like promoters during a diauxic shift served as an additional test of our computational approach, i.e. our ability to predict global regulation purely from the growth rate. To provide further validation of our approach, we tested our

ability to predict promoter activity in 6 additional perturbations which do not constitute carbon source changes and which were not part of the original data set. As expected given the dominance of global regulation that we identified in our study, the vast majority of central metabolic promoters were indeed well predicted by our approach based on growth rate alone. The exception was growth at 42°C, possibly due to temperature-dependent effects on global regulation. Thus, we conclude that global regulation has a strong impact on promoter activity also beyond changes in carbon sources, and that we are able to robustly quantify and predict its impact on promoter activity. We included these data in the revised manuscript (supplementary figure 8).

4. Some reviewers believe the conceptual novelty to be limited

Both reviewers #1 and #2 argue our work to be mostly confirmatory. We agree that we are not the first to attempt the dissection of global and specific regulation. However, we argue that our approach and conclusions represent a significant improvement over the three studies previously published in MSB. The approaches by Gerosa et al and Berthomieux et al require the construction of constitutive promoters to quantify growth rate dependent changes in global regulation, which might not always be feasible, whereas our approach does not require dedicated constitutive promoters for normalization. The study by Keren et al is conceptually similar to our work, but was restricted to pairwise comparisons of conditions. Most importantly, however, neither of these previous approaches attempted to systematically identify the regulatory signals (i.e. metabolites) underlying specific transcriptional regulation. In our view, the identification of such regulatory signals is a major challenge when studying the interplay of metabolism and cellular regulation. To our knowledge, the computational inference approach presented here constitutes one of the first efforts to tackle this challenge for a biologically relevant regulatory network, and is also easily scalable and widely applicable (the only requirements are gene expression and metabolite data in matching conditions). A major value of our approach is its relative simplicity that does not require a complex model, which renders it amenable to a wider application in the community. The value of this approach was also acknowledged by reviewer #3.

Another novelty concern raised by reviewers #1 and #2 regards our obtained results and the conclusion that essentially two transcription factors, Cra and Crp, explain most of the observed specific regulation in central metabolism. In particular reviewer #1 claims this observation to be fully expected given previous modelling efforts from Kotte et al. Here, we respectfully disagree: the modelling approach by Kotte et al merely ASSUMED that four specific transcription factors are relevant to regulate metabolism in one particular diauxic shift (from glucose to acetate), but did not actually prove it. In stark contrast, we start from a quite extensive data set and find by analysis two transcription factors sufficient to quantitatively explain transcription of nearly all individual promoters in central metabolism across a wide range of conditions. Both in scope and conclusion these are, in our view, very different things. In our view it is one thing to make generic statements such as “Cra regulates glycolysis” but quite another to provide quantitative evidence which factors do precisely what and specifying to which extent a factor explains the regulation.

Notably, our data-driven approach reveals several marked differences in regulatory topology to the ASSUMED model of Kotte et al (e.g. regulation of *pfkA* and *pckA*). We believe that data and conclusions such as presented here may in fact guide future computational efforts (such as the one by Kotte et al) to construct general kinetic models of central metabolism that only include those interactions that actually matter *in vivo*. In fact, the surprise of reviewer #2 (“why does *E. coli* have the other 30 TFs then?”) – who is certainly an expert in the field – suggests that our conclusions are likely going to be non-obvious to the broad readership of MSB as well.

Finally, reviewer #2 claims that the importance of Crp regulation for central metabolism was expected due to a link between the ATP/ADP ratio and the growth rate. We would like to respectfully point to recent work from the lab of Terence Hwa, which convincingly showed that Crp activity is largely determined by the balance of catabolic activity and anabolic capacity, mediated by the allosteric inhibition of adenylate cyclase by keto-acids (You et al., 2013. Nature). In fact, our data shows that the ATP/ADP ratio remains constant across the tested conditions, whereas the concentration of the main keto-acid 2-oxoglutarate does exhibit the expected trend when comparing the carbon source conditions and chloramphenicol treatment (see supplementary figure S9). Thus, our results not only confirm the reported observations from the Hwa lab, but also show specifically which promoters actually follow this regulatory regime.

“Radically original or novel” (reviewer #2) is a tough call, but we hope to have given sufficient arguments on the novelty of our findings and the originality of the approach (none of the reviewers have pointed to a paper that precedes our analysis). We strongly feel that the claim for lack of novelty is unfair because nobody has been able to demonstrate (or known) that only 3 metabolic

signals via 2 transcription factors plus the global growth rate regulation suffices to explain expression of all CCM genes under more than 20 conditions. Sure the two factors are not a surprise as such, how could they given the huge amount of data on *E. coli* (!), but providing evidence that they are all that it takes is entirely novel. Whether or not this is "radically" novel or not for reviewer 2 is another matter.

In the revised manuscript, we expanded the discussion to better explain the novel aspects of our work. In particular, we made the main achievements of this study – to demonstrate quantitatively which factors explain transcription in central metabolism, and to reveal the metabolic input signals – more clear in the discussion (i.e. in L372-380). Finally, we made clearer that our approach is widely applicable to any other microbial system for which static metabolomics and transcript data are available (L415-416).

5. Reviewer 3 asked us to explain our computational approach to quantify the contribution of global regulation in more detail.

We expanded the methods section of the manuscript to better explain the individual steps of this approach (L480-488), and re-phrased the respective paragraph in the main text (L149-174).

Specific comments:

Reviewer 1:

In summary: The results presented here are a follow-up of previous work. At least three previous publications in MSB (Berthoumieux et al., 2013, Gerosa et al., 2013, and Keren et al., 2013), all cited in the present manuscript, have demonstrated that global regulation dominates over specific regulation. The first reference used detailed dynamical analysis on a limited set of promoters; the second, by the group submitting the present manuscript, shows growth-rate dependence of global regulation in steady state as well as dynamical transitions on a larger set of genes; the third reference shows a similar result on a genome wide-level at steady state in different growth conditions. The present work thus confirms these previous publications for other promoters and slightly different experimental conditions.

The measurement of metabolites in the different growth conditions adds interesting information about the metabolic signals that are responsible for specific regulation of genes involved in central metabolism. The major conclusion of these measurements is the observation that very few metabolites, essentially cAMP and fructose-bis-phosphate (FBP) account for most of the observed, specific regulation. The dominant effect of cAMP and FBP (and two other metabolites) had been established in 2010 by a modeling approach, also published in MSB (Kotte et al., 2010).

The work presented in this manuscript thus confirms the conclusions of four previous publications in MSB and does therefore not add much novel insights or conceptual advances. In other words, the manuscript lacks originality and novelty. However, the information presented is still useful and the experiments, as well as the analysis of the results, are well carried out.

We disagree entirely with this assessment. The main focus of our work is not to confirm that much of the expression in central metabolism is based on global regulation, and we believe it is simply not correct that state that the Kotte paper established the dominant effect of cAMP and FBP. This paper is a great piece of work, but it assumes a regulatory logic while in our case we infer the regulatory logic from data in an unbiased manner. We would also like to point out that our approach is, in our view, quite original and useful beyond the system studied, hence there is also value in the method as such. Please see the general rebuttal response 4 for a more detailed answer.

Specific comments:

1. The authors insist that they measure transcriptional regulation. However, since they use fluorescent reporter genes to assess promoter activity, some global effects could also be mediated by changes in translation efficiency.

This is an important point, please see our response to common main concern 1.

2. The reporter plasmids used are taken from the promoter collection of Zaslavler et al. (2006). These constructions are not strictly speaking transcriptional fusions since they contain part of coding sequence of the gene that is replaced by *gfp*. Again, this could lead to regulatory effects other than purely transcription. Furthermore, a number of reporter plasmids have been constructed in this study. The sequences of these constructs should be given in the SI.

Regarding the concern about regulatory effects other than transcription, we refer again to our general rebuttal response concern 1. Regarding the promoter construction: The reviewer is right about the way the promoter collection was constructed. This approach is motivated by the fact that in some cases the binding sites of transcriptional regulators can extend into to coding sequence of the gene (Zaslaver et al., 2006, Nat Methods). Information on the newly constructed promoters will be provided as part of the supplementary material.

3. The key parameter of the study is growth rate which was estimated from a "time-range visually identified as exponential phase". A simple figure in the supplementary material should show one example to illustrate the procedure.

Here we used the same approach as the study by Keren et al, 2013, MSB. We included an example figure in the supplementary material (supplementary figure S1).

4. The observation that parameters obtained from steady-state measurements can predict dynamical transitions (Figure 2D) should be commented. This result implies that changes in global transcriptional regulation are fact compared to the time-scale of a diauxic shift.

The reviewer is right, in fact in our previous work on the arginine biosynthesis pathway (Gerosa et al, 2013, MSB) we had made the same observation using a more mechanistic kinetic model of promoter activity (figure 2 in this publication).

Reviewer 2:

Summary. Kochanowski et al. present detailed analysis of regulation of ~100 genes of central metabolism in *E. coli* by 30 TFs in 26 environmental conditions. They constructed a library of promoter-GFP constructs for each gene and investigated the activity of the promoters by assaying GFP fluorescence during growth in media that differ in nutritional composition (primarily C-source). They go on to explain differential expression of genes as a function of global and specific regulation. With respect to the latter, they conduct metabolite profiling to elucidate which metabolites are responsible for the condition-specific regulation of promoters whose activity could not be explained entirely by global regulation, which their data suggest is directly correlated to growth-rate and attributable to regulation by just two transcription factors (CRP and Cra). The metabolite profiling implicated cAMP, fructose-1-phosphate and fructose-1,6-biphosphate as major influencers of transcriptional regulation of central metabolism genes. Based on the results from this study they claim that few inputs regulate the transcriptional response of central metabolism in *E. coli*.

Review. Generally speaking, the manuscript is well written and easy to read. There were some glaring omissions in the introduction and discussion, which need to be addressed. For instance, work on systems biology of transcriptional regulation in *E. coli* is largely ignored -they should fix it to give a more unbiased view of prior work. But otherwise, the intro was well written and it introduces the necessary concepts to understand the presented results. The Methods section is well-written and detailed. Translation of different biological influences into a mathematical formalism was done elegantly. I also enjoyed reading the precise and robust justification for the choice of different thresholds used in the study (section 2 of Sup. Info.). Additionally, some negative results are also communicated which is really appreciated. However, I did not find the general approach of the work to be radically original or novel. I also have major concerns about the conclusions -especially, I find the evidence to be insufficient to support the claims. Even if the claims were supported, I am not particularly convinced that the insights are novel -in the limited context in which they might be relevant (i.e., diauxic shift).

We are grateful for the constructive and generally positive comments. The concern on "radical" novelty of our approach and the novelty of the insights was addressed in the general rebuttal comment 4. Regarding the claim that we omitted to cite relevant work in our introduction: we already cite a variety of recent high-profile studies which have focused on transcriptional regulation in *E. coli* (and other microbes). Regretfully, the reviewer didn't specify which relevant works we omitted, but we will of course be happy to include them if the reviewer points them out to us.

The authors make a big claim that a significant proportion of transcriptional regulation of most central metabolism promoters is explained by growth rate -a claim that needs to be supported by compelling evidence. For reasons described below, more experiments are needed to support that claim. A better experiment design would be to take away the effect of growth rate by growing

cultures in chemostat mode, and then evaluate the consequence of a shift in C-source. This is admittedly a complex experiment design that is not easily scalable to many promoters -but I would like to see it done for at least few promoter-reporter construct strains. In particular, I would like to see how temporal changes in transcript levels of some of the 100 selected promoters correlate to changes in GFP fluorescence during a shift in C-source. The chemostat experiment design will help to address this major question that is central to all of the conclusions and claims: does turnover dynamics of GFP capture subtle changes at the level of transcriptional regulation? This is especially a concern for genes that experience transient up or down regulation at the transcript level that is unlikely to appear at the GFP level. Hence, it is not surprising that majority of regulation is correlated to growth rate -since protein synthesis of GFP will depend on abundance and availability of ribosomes. By contrast, subtle changes at the level of transcriptional regulation will be somewhat masked by the dominant influence of growth rate-correlated protein synthesis.

We are not the first to point out that a major fraction of transcript changes in central metabolism is primarily a function of the growth rate. It is a very important point though whether our data would be confounded by the choice of experimental set up (ie batch versus chemostat cultures). We believe that this is not the case and we added new data. The whole argument is outlined in detail in the general rebuttal comment 3.

It is also not surprising that cAMP, Fructose-1,6-biphosphate, and fructose-1-phosphate explain gene regulation of most genes of C-metabolism. Whereas consumption of the two phosphorylated sugar derivatives would be expected to reflect C-uptake rates through glycolysis, the level of cAMP is a reflection of the overall energy status (ATP/ADP ratio) of the cell -both of which are directly proportional to growth rate. These issues worry me that the observations are confounded by the flawed experiment design that expectedly would have catabolite repression and growth rate dynamics as dominant factors that mask all other transcriptional inputs. The claims are bound to be true only in the specific context of diauxic shift from one C-source to another -in which case, the results are not surprising and something that has been known for quite some time.

We disagree with this assessment. First, a key conclusion of our work, derived directly from experimental data, is that 3 metabolic signals via 2 transcription factors plus the global growth rate regulation suffice to explain expression of all CCM genes under more than 20 conditions. These conditions cover a variety of glycolytic and gluconeogenic carbon sources, as well as direct perturbations of the global expression machinery, and it is far from obvious that so few regulatory signals would be sufficient to coordinate CCM gene expression in all of these cases. Previous works, such as the one by Kotte et al, 2010, MSB, had simply assumed that few regulatory signals may be sufficient, here we show it directly. Second, we disagree that our findings are confined to diauxic shifts: in fact, the chloramphenicol treatment conditions that we used here are clearly not carbon source changes. Moreover, data on 6 additional conditions that had not been included in the original manuscript suggest that global regulation plays an important role beyond carbon source changes (see also our response to main common concern 3). We outline the whole argument in detail in the general rebuttal comment 5.

Another point that was surprisingly not discussed was the generally accepted view that regulation is not just about turning up or down genes independent of each other, rather it has to do with coordinating sets of genes that carry out related functions in relevant environmental conditions. Their claim is extraordinary in that "...a surprisingly simple regulatory program that relies on global transcriptional regulation and input from few intracellular metabolites appears to be sufficient to coordinate E. coli central metabolism, and explain about 90% of the experimentally observed transcription changes in 100 genes." I am especially curious to know why there are 30 transcription factors maintained in the E. coli genome - all apparently implicated in regulation of central metabolism genes -if only two transcription factors are sufficient. The authors hand-wave that these transcription factor might function in other stress conditions. This statement could be investigated further by mining the extensive compendia of publicly available transcriptomes for E. coli. In that regard, it was puzzling why other bodies of work that have looked at conditional operation of the E. coli gene regulatory network from a systems biology perspective were notably omitted from the introduction and discussion.

We argue that the regulatory mechanisms what we identify in this work do in fact constitute a program that ensures the coordinated expression of CCM genes in various conditions (as written in the abstract, as well as the discussion, e.g. in L384-385). Regarding the biological role of the other transcription factors: we maintain that these are likely to be relevant in other conditions that we did not test here. However, proving that they are NOT relevant in the tested conditions is obviously

difficult. The reviewer suggests to address this apparent conundrum by tapping into the wealth of already published transcriptomics data. While we do appreciate this suggestion, we would like to point out that the vast majority of large-scale transcriptomics data sets do not include the crucial growth rate information (e.g. the commonly used M3D reference data set from Faith et al., 2008. Nucl Ac Res). Moreover, the data shown in supplementary figure S8 already provide anecdotal evidence that additional transcription factors may be relevant when considering further conditions. For example, the two promoters whose activity deviates the most from the growth-dependent prediction upon treatment with the oxidative stress inducing agent paraquat are *zwf* and *pgi*, both of which are activated by the oxidative stress regulator SoxS. We included this additional evidence in the revised manuscript (supplementary figure S8).

The reviewer also again refers to apparently relevant studies that we did not cite in our work. As we already wrote in our response to this reviewer's first concern, we cited many relevant and recent studies on the systems biology of transcriptional regulation in microbes, and are not aware of glaring omissions. But we will of course include further relevant studies if the reviewer points them out to us.

1. The number of conditions they list (26) is a bit overstated, as this number also includes control conditions in which cultures were grown with chloramphenicol (based on my count the correct number should be 17 different growth conditions).

The chloramphenicol treatment experiments are not mere controls, but an integral part of the data set which enables us to study the effect of growth rate on promoter activity independently from carbon source changes. Here, we follow previous works (e.g. Nichols et al, 2011. Cell. PMID 21185072) which treat different chloramphenicol concentrations as separate conditions.

2. At several places, (L43, L48, L317), authors use the term "adaptation" when I assume they meant "acclimation". Authors refer to the process by which cells adjust to a gradual change in its environment not to the dynamic evolutionary process by which a trait evolves in species populations. Authors should address this confusion

Indeed we investigate cellular response to environmental changes. We understand that the term adaptation is often used in an environmental context, but we disagree that its meaning is exclusive to an evolutionary context. Since most papers use "adaptation" in a very general sense (e.g. Nicolas et al, 2010. Science; Kao et al, 2005. JBC; Kotte et al, 2010, MSB to name but a few) we prefer to stick to the original terminology.

3. Figures 1, 2b and 5 specifically highlight the condition "5 mM cAMP". Why? What is the rationale? Further background about the importance of that condition is expected. Please elaborate.

The 5 mM cAMP condition is an orthogonal perturbation (i.e. external activation of a transcription factor, in this case Crp) to the carbon source/chloramphenicol conditions and was therefore highlighted separately. Moreover, it also is a further validation to identify promoters whose activity depends on Crp (see e.g. *talA*, *acs*, *sdhC* promoters in figure 1).

4. Paragraph L343-357 is generally difficult to understand. Please rewrite for clarity. Specify the major point and provide with sufficient background such that a broader audience will understand the message conveyed and the relevance of the finding.

We re-wrote this paragraph to (main text L378ff):

"The effect of Cra on glycolytic promoters was rather weak, only modulating the dominant global transcriptional regulation. What could be the physiological relevance of such a modulating regulatory signal? One attractive hypothesis emerges when considering protein concentration (as the final output of gene expression) (figure 5). Proteins that are expressed by constitutive promoters show a negative relationship between growth rate and protein concentration when varying carbon source availability, see also (Klumpp et al, 2009). If such proteins are required at high concentration during fast growth (i.e. to carry a high metabolic flux), global regulation alone will lead to even higher concentrations at slow growth (figure 5B, upper panel), putting additional burden on the cell. Since glycolytic carbon sources tend to support fast growth and result in higher FBP concentrations, regulation through Cra-FBP (i.e. repression by Cra which is alleviated by FBP) may counter this effect, causing more constant protein concentrations across different growth rates (figure 5, middle panel). Conversely, the regulatory input of Crp-cAMP yields a previously described linear negative relationship between protein concentration and growth rate under carbon limitation for catabolic proteins (You et al, 2013; Hui et al, 2015) (figure 5, lower panel). Thus, few

such regulatory signals may allow cells to coarsely allocate proteome resources based on the supported growth rate."

5. L378-L380. Last sentence is irrelevant. Please remove. A conclusion tone sentence which summarizes the work and its relevance is advised.

We have already concluded on the scientific findings in the previous paragraph. In this final paragraph we rather prefer to end with a sort of outlook how the presented generally applicable approach can also be applied to other regulatory networks. In our view, this an important aspect of this manuscript.

6. In the introduction, I would recommend to discuss the role of promoters in the regulation of metabolism.

We included additional references to additional studies that focus on the role of promoters in the regulation of i.e. uptake systems (for which most of the data is available).

7. L103-L105. I recommend to improve the readability of the sentence. How many reporter plasmids were used initially? At first glance, the relationship between the genes and the promoters was not clear.

We modified the sentence. Specifically we used 95 promoters in total.

8. L126-L128. Hard to understand. What is the message of the sentence? Please rephrase.

Here, we wanted to highlight the observation that the tested synthetic constitutive promoters have very similar activity patterns as many of the tested promoters of central metabolism. We re-phrased this sentence for clarity.

9. L261. If PEP is a false positive, are all others with lower target overlap also false positives? Why? Please discuss or elaborate.

We restricted our analysis to those transcription factors that show the largest target overlap with the respective metabolite. If the target overlap is poor, our analysis is unlikely to be conclusive (e.g. for DcuR, which shares two target promoters with PEP), since in this case hypergeometric testing is prone to favor transcription factors with few (or only one) targets. We clarified this aspect in the respective methods section.

10. L311-314. It's not clear to which 2 TFs and 3 metabolites authors refer to. Please name them for clarity.

We added this information.

11. Syntax used in L449 ("promoter- and") and L456 ("promoter- and") is unclear. Please rewrite sentences for clarity.

Done.

12. L486. Based on what criteria are those 12 promoters selected. Please explain.

We manually selected promoters with diverse activity patterns (i.e. constitutive-like, activation in few conditions, more complex activity patterns), which we also explain in the text.

13. Figure 4. Some of the labels touch the grid lines, please fix.

Fixed.

14. L31. A comma is missing between "Remarkably" and "cyclic".

Fixed.

15. Eq. 2. Second term, S. At the end of equation, subindex of TF should be lj, not li.

Fixed.

16. L268 of Supp. Info. There should be a space between value and units in "2g/L".

Fixed.

Reviewer 3:

This study investigates the regulation of expression, both global and specific in central carbon metabolism. It shows the dominant role of the global response in central carbon metabolism. One of the impressive features of the study is the ability to systematically find metabolites that regulate

transcription factors activity. I found the work timely and useful. My suggestions for improvement are detailed below.

We are grateful for the positive assessment and the constructive comments.

Major points:

1. L. 152-154: This is where I think most readers will lose the ability to follow what was done. "When applying singular value decomposition on 153 the log normalized promoter activity data, few common patterns, or singular vectors, captured most 154 of the data set's variability (supplementary figure 4)." SVD is not a simple thing and an intuitive explanation will be useful. It took this referee a long time to understand the axis, and so probably also to other people who will aim to really follow the details of what you did. What are then the relevant "axis" of the PCA/SVD? You should take the reader by the hand on this very challenging turf.

Thanks for the advice. We took care of this point, please see general rebuttal comment 5.

2. The authors find that the first and quite dominant SVD axis is strongly related to growth rate. They state: "As hypothesized, the first singular vector showed strong growth rate dependence (figure 2B)." I think this is a key point and several things should be done to clarify it to the reader.

- a. take the reader by the hand in explaining what are the steps done to achieve this result. Is it a correlation of the values in the 1st SVD direction of each condition point to its growth rate? This requires explanation at least to this reviewer.

Again, thanks for the advice. We took care of this point, please see general rebuttal comment 5.

- b. What is the shape of the "dependence"? Can you plot it? You mention also doing a z transformation to the log PA but is the dependence with or without the z transform?

We plot the relationship between growth rate and first singular vector in main figure 2B. The z-transformed log PA data were the basis of this plot.

- c. The slope of the correlation before the z transform seems to me to be quite informative (unless I am misunderstanding, which frankly could be the case), if the slope of the 1st SVD axis to the growth rate is close to 2 it might suggest an underlying model. If it is quite different than 2 that is also quite interesting to point out somewhere. I might be off here.

Unfortunately we do not fully understand the reviewer's argumentation here, as it is not clear to us how the slope of the relationship between the first singular vector and the growth rate can be used to suggest an underlying model (i.e. it is unclear what the reviewer means by "model" in this context).

3. This study has a wealth of data that I would imagine some researchers would like to analyze further with their own hypothesis. It will be good if the authors publicly provide the raw data measured as well as the processed data and the code for the data analysis.

This was our intention, and has always been our style. Of course we will include the relevant files in the supplementary material.

4. Figure 1/S3: The most clear distinction that I saw was between the chloramphenicol treatment conditions and all the rest. I feel this not so common condition swamped all the other things and was too bad given my interest in the "normal" growth conditions. I would urge the authors to also make a plot without the chl treatments that will appear in the SI if not the text. If indeed this has an effect as seems to me by eye, this issue should be pointed out and discussed in the text explicitly.
5. Figure 2A: Much of the explanatory power of the SV1 seems to come from the chloramphenicol conditions according to my understanding of the figure. What would be the results without these conditions? The authors should analyze and at least reflect and mention explicitly this point.

A very reasonable concern, but we can assure the reviewer that there is no major bias introduced by the chloramphenicol condition. We explain this now in the text. Please see general rebuttal response 2 for a full answer.

6. Around L. 200: Can you give a value of how much of the variability in expression that is specific you could explain with the top 1, 2, 3, 4 metabolites modulators?

Calculating such values directly is difficult since for some promoters two metabolites are identified as modulators. However, given that the top modulator, cAMP, accounts for approx. half of the

detected promoter-metabolite interactions, we estimate that it is responsible for much of the explanatory power of the metabolite inputs.

7. 330-332: "In particular FBP/F1P inhibition of Cra regulates glycolysis and cyclic AMP activation of Crp regulates expression of TCA cycle and carbon source utilization pathways." This nice view would gain a lot in my eyes from having a figure associated with it that shows the model of regulation of central carbon metabolism proposed by the authors in one simple schematic as the discussion suggests. I think this can become textbook material if done properly, and I would surely find it useful personally for me and my students.

We included a corresponding figure in the supplementary material (supplementary figure S19) and hope that this figure can be useful to the reviewer (and of course to all other readers of the manuscript).

8. It is not clear to me if the global response should be thought of as "global regulation" or as a passive response that is an outcome of resource allocation and thus is not an active act of "regulation". My personal view is the latter but in any case I think the reader can gain from at least mentioning this distinction in the discussion.

Here we use the term "regulation" operationally as "something that changes the activity of a promoter". The exact mechanism underlying global regulation, and its relationship with resource allocation, are still a matter of active research, and we argue that a detailed discussion of these aspects is beyond the scope of this manuscript.

Minor points:

9. In lines 138-140 where you introduce the functional form of eq. 1 the reader can be perplexed about why there is the alpha exponents. It will be useful to explain that this is not for mechanistic reasons (as far as I understand) but mostly driven by mathematical convenience for the next step where everything will be analyzed in log space and thus this will lead to a solvable linear relationship.

Indeed, here we use power law terms to approximate the non-linear effect of expression machinery and transcription factor activity on promoter activity. We made this point clear in the detailed description of the mathematical model (supplementary text 1).

10. 144, eq.2, the $\Delta(\log(p_{ij}))$ is not defined so clearly. I would write as $\Delta(\log(p_{ij})) = \log(p_{ij}) - \log(p_{i0}) = G + S$. Also in Eq.2 the \sim relation for S is far from being immediately clear. Need to explain.

The detailed explanations are given in supplementary text 1.

11. L 27-28 "approximating each term expressed as $\log(1+x)$ with $\log(x)$ " should write when is this reasonable ?

This approximation is reasonable if $x \gg 1$. In the context of our work, this would be the case if a transcription factor strongly influences a promoter. Nevertheless, in our simulations (supplementary text 2), where we used the same approximation, we saw that even for weak transcription factor mediated regulation (summary of simulations figure A, right panel) our approach is still able to detect the vast majority of promoter-metabolite interactions. Therefore, we conclude that this approximation has little effect on our ability to promoter-metabolite interactions.

12. L 31-32 "assuming that transcription factor expression does not change significantly across conditions". This is a very strong assumption. I do not know how well it holds but in any case the fact that you are using it should be made more explicit in text rather than just in the SI.

This assumption is mentioned and justified in the main text (lines 198 to 202).

13. Many readers have heard of PCA but not SVD. It will be good to more clearly state their intimate relationship (synonyms for the purpose of this paper?).

For the normalized data used on this manuscript, PCA and SVD yield identical results: the first principal component and the first singular vector will have exactly the same values. We discuss this aspect in the revised explanation of the computational approach, see our response to main common concern 5.

14. Equation 2: shouldn't S, G also have subscripts of i/j ?
To not overburden the equations, we omitted i/j in these cases, which should be evident from the definitions of S and G in equation 2.
15. Equation 3: alpha is defined above equation with one index and in equation with two. It will also be good to explain again what i and j are in this context.
Equations 2 and 3 contain the same alpha term. We added a pointer to the definition of i and j .
16. 217-218: "In total, for about 50% of the promoters at least one of the available metabolites could explain the specific transcription as a single input." How do the authors define "could explain"? is there a numerical definition?
"Could explain" is defined as: about 50% of the promoters had at least one metabolite with a correlation coefficient of >0.75 or < 0.75 (see supplementary figure S12).
17. 232-233: "and scored the improvement in agreement between measured and predicted promoter activity over global transcriptional regulation alone (figure 4A)." I do not think this is shown in 4A.
Sorry, this sentence is a remnant from a previous version of the manuscript, we removed it.
18. 270-272: "These results highlight the importance of taking global regulation into account when interpreting promoter activity measurements, especially when the growth rate is different between strains or conditions." I liked this observation which I think is very insightful. The authors may want to mention a recent approach (PMID:27073913) that highlighted the utility of using the growth rate dependent global response as a null model on which to detect specific responses.
We thank the reviewer for pointing out this reference, we included it in the main text.
19. 275-276: "(such as between *ppc* and *Cra* (Shimada et al, 2010), or between *pgi* and *Crp* (Shimada et al, 2011))", From the text I thought you are talking about promoter-metabolite interactions but the examples are between TF and promoter it seems. Please clarify.
In this section of the manuscript, we move our attention to the transcription factors which ultimately mediate the detected promoter-metabolite interactions (figure 4A/B), in this case *Cra*-FBP/F1P and *Crp*-cAMP. We added this information for clarity.
20. 284-286: "Reassuringly, we did not detect any examples of reported promoter metabolite interactions that were only recovered if the respective promoter's global regulation component had not been removed." There are double or triple negative here that was hard for me to parse.
We re-phrased this sentence.
21. 313-314: "was sufficient to explain the majority of changes in promoter activity across conditions." How much is the majority? Will be useful to have a number
Here, we refer to figure 4C. From the R^2 value between measured and predicted promoter activity, we conclude that $>90\%$ of the variability in promoter activity can be accounted for by global regulation plus few regulatory metabolites. We add another pointer to figure 4 in the text.
22. 331 "Cra regulates glycolysis and cyclic AMP activation of Crp regulates expression of TCA cycle and carbon source utilization pathways." Can the word "regulates" be changed to something more concrete like up-regulates or down-regulates?
Since we find *Cra* represses glycolytic promoters and activates gluconeogenic ones, we decided to use the generic term "regulates" for the sake of brevity.
23. 346-348: "For promoters that are solely subject to global transcriptional regulation, variation of carbon source availability yields a negative relationship between growth rate and protein concentration." I see this in the figure (5B??) but what is the reason for this? Was this predicted from your model? Even if not, that is worth stating explicitly for the benefit of the readers so they know what is understood and what is not understood at this time.
This relationship between growth rate and protein concentration is obtained when dividing promoter activity by the respective growth rate (in steady state, see also Gerosa, Kochanowski et al, 2013).

MSB). We note that the same negative relationship for constitutive promoters emerges from theoretical considerations in Klumpp 2008, Cell. (figure 2D).

24. 369: "for the broad range of here investigated" typo
Fixed.

25. 435-440: It will be useful to say how many promoters were tested and how many could be reliably quantified above the background on average across the different conditions. We tested 95 promoters, 31 of which were inactive (= below/around background) in all tested conditions. We modified this section for clarity.

26. 441-445: For the SVD, what are the dimensions of the space? Help the reader by explaining explicitly if the axis are promoter activities of each gene and the points are the conditions, or the axis are the promoter activities in each condition and the points are promoters. It is all trivial to the authors I imagine but for the readers it can be challenging.
See our general rebuttal response 5.

27. Figures 1-5: the red/blue/green data points often overlap very much. I think that a bit smaller points would help the presentation. Removing the outer black lining around them would probably also help discern the colors.
We decided for this presentation to ensure that these figures are still readable when printing the manuscript with two pages per sheet.

28. Figure 2: The color red means different things in panels B, C, D. You might want to update this. A similar things happens in Figures 3 and 4.
We adapted the color code in figure 2. For the other figures, we decided to keep the number of colors per plot as low as possible, in the hope that the prominent accompanying legends should make the distinction clear to the reader.

29. I did not find an overall description of the variability in promoter activity, what is the average CV of a promoter across conditions? does it depend on the expression level?
Our median day-to-day variation of promoter activity is below 15%, and only weakly depends on the promoter activity (see supplementary figure S3).

30. Comparison between the promoter activities and other data sets, where data already exists (e.g. Schmidt+Heinemann proteomics) can be of interest.
See our general rebuttal response 1.

31. If I understood correctly, the global effect is opposite to that predicted by Klumpp 2009. This is worth pointing out explicitly.
The global effect on promoter activity is quite comparable between our study and the one by Klumpp 2009 (i.e. transcription rate in figure 1A of Klumpp 2009 and figure 2B our manuscript).

32. It is interesting that 70% of the variability is explained by growth rate where there is usually 10-20% of noise in such measurements. Might be good to mention something about noise level.
See our answer above to comment 31 of reviewer 3.

33. Figure S5 caption: state reference to full data. Mention the basis of the log is 10.
We used the natural log in this plot, and added this information in the plot.

Thank you again for submitting your work to Molecular Systems Biology. We have now heard back from the referee (reviewer #1 of the previously rejected manuscript MSB-16-7141) who agreed to evaluate your study. Reviewer #2 of the previous submission was not available for evaluating the revised and extended version of the study. As you will see below, reviewer #1 thinks that the study has been improved and the main issues raised have been satisfactorily addressed. However, s/he lists a few remaining issues, which we would ask you to address in a minor revision.

In line with the reviewer's comment on data documentation, we would ask you to include a Data Availability section at the end of the Materials and Methods describing where the newly generated data are available (i.e. as EV Datasets or in an appropriate database).

REFEREE REPORTS

Reviewer #1:

The revised version of the manuscript adds some additional information and explanation, and the rebuttal by the authors clarifies some of the issues.

First of all, thank you for taking care of all the detailed remarks by myself (few) and by the other referees (many more). I will focus on the major points, also addressed as such in the rebuttal, and add just one additional request for clarification and original data (in point 1).

1. Transcription versus translation

The authors explain in their answer (and the data of Gerosa, 2015, show) that the effect of specific regulation is small: generally less than 2-fold. Since almost all data are presented as z-score normalized plots, we have no access to the absolute effects. You should clearly state somewhere in the main manuscript that we are dealing with small effects. If all the effects are small, why did you limit your analysis of Figs S16 and S17 to genes that contain at least one data point with a value greater than 2-fold?

Please show absolute values of promoter activities in the supplementary information. Please include error bars as well, since you can estimate the error from data as the ones in Figure S1 by just taking the range of promoter activities within the shaded region. An error estimate is all the more important since the example shown in most of the figures (and the previous publications of the group), *pykF*, does not correlate well with the transcriptomics data (Fig S17).

There is another small concern about quantification. The largest SV (a proxy for growth rate or global regulation) was used to predict promoter activity during the diauxic shift from glucose to succinate (Fig2D). The authors state line 176: "... found that promoters whose steady state activity was dominated by the first SV were also well predicted during the diauxic shift." Already the third ranked promoter, *pfkA*, seems to be an exception. You may want to comment on this.

2. Dominance of chloramphenicol conditions

I had missed this point in the first round of reviews. The observation could be emphasized even more prominently in the manuscript.

3. Global regulation and different carbon sources

I agree that the major control variable is growth rate. However, this conclusion is not novel.

4. My major argument concerning the novelty and originality of the manuscript was based on extrapolation of previous work on global regulation and a MODEL (as I said) showing that four sensors are sufficient to explain the metabolic adaptation during a diauxic shift from glucose to acetate. I fully agree that the Kotte publication is modeling work. Nevertheless, some experimental evidence the authors are certainly aware of has been added later. The first author of the present manuscript is also first author of the PNAS publication showing experimental evidence for one of the flux sensors, *FBP*.

I respectfully apologize and admit that my initial judgment may have been a little harsh. The conclusions of the present work were expected (in my opinion). The major merit of the manuscript is to EXPERIMENTALLY SHOW that this expectation was warranted. I agree with this claim.

5. OK

In summary: The revised manuscript is improved in many respects. The conclusions are not surprising. The major virtue of the work is to experimentally demonstrate that two transcription factors (and essentially two metabolites, considering *FBP* and *FIP* as closely related) control most of the specific expression of genes involved in central carbon metabolism. Since the major strength of the work is experimental, all of the original, raw data (and error estimates) should be provided in supplementary files.

Point-by-point response**Reviewer #1:**

The revised version of the manuscript adds some additional information and explanation, and the rebuttal by the authors clarifies some of the issues.

First of all, thank you for taking care of all the detailed remarks by myself (few) and by the other referees (many more). I will focus on the major points, also addressed as such in the rebuttal, and add just one additional request for clarification and original data (in point 1).

Thanks for the willingness to look at this MS again and the constructive comments.

1. Transcription versus translation

The authors explain in their answer (and the data of Gerosa, 2015, show) that the effect of specific regulation is small: generally less than 2-fold. Since almost all data are presented as z-score normalized plots, we have no access to the absolute effects. You should clearly state somewhere in the main manuscript that we are dealing with small effects.

When averaged over all changes, the effect of specific regulation is indeed rather small, accounting for 32% of variability in our data set as described in the main text (L199-200). However, for individual promoters the effect of specific regulation can be substantial, or even dominate the promoter activity response, as we show in main figures 2C and 4A. In nearly all such cases with strong specific regulation the promoter activity faithfully recapitulates the transcriptomics data (Appendix Figure S17), supporting our argument that we are not observing effects that are specific to the promoter activity. We clarified this aspect in the manuscript (L259-262).

If all the effects are small, why did you limit your analysis of Figs S16 and S17 to genes that contain at least one data point with a value greater than 2-fold?

In our comparison of transcript levels and promoter activity, we follow the conventions of transcriptomics studies, which frequently use a 2-fold change as an empirical cut-off to identify significant changes. For smaller changes, such comparison is unlikely to give conclusive results given the experimental uncertainties of transcriptomics and promoter activity measurements.

*Please show absolute values of promoter activities in the supplementary information. Please include error bars as well, since you can estimate the error from data as the ones in Figure S1 by just taking the range of promoter activities within the shaded region. An error estimate is all the more important since the example shown in most of the figures (and the previous publications of the group), *pykF*, does not correlate well with the transcriptomics data (Fig S17).*

The measured promoter activities were included as supplementary information (EV table 3), together with error estimates based on day-to-day variability measurements (appendix figure S3) following the approach by Keren et al, 2013, MSB.

*There is another small concern about quantification. The largest SV (a proxy for growth rate or global regulation) was used to predict promoter activity during the diauxic shift from glucose to succinate (Fig2D). The authors state line 176: "... found that promoters whose steady state activity was dominated by the first SV were also well predicted during the diauxic shift." Already the third ranked promoter, *pfkA*, seems to be an exception. You may want to comment on this.*

The reviewer is right in pointing out that not all of the promoters whose steady state response is dominated by global regulation can be well predicted during the diauxic shift, in particular during the lag phase. One possible explanation of this finding are regulatory events which do not occur during steady state growth (and were therefore not detected by our approach). For example, *pfkA* (as well as *pykF* and *rpoH*, two additional promoters which show measured deviations from the promoter activity predictions) is subject to regulation by the stress sigma factor RpoS, which is presumably only active during the lag phase. We included a comment on this finding in the main text (L181-184).

2. Dominance of chloramphenicol conditions

I had missed this point in the first round of reviews. The observation could be emphasized even more prominently in the manuscript.

We re-phrased the passage for clarity (L165-167).

3. *Global regulation and different carbon sources*

I agree that the major control variable is growth rate. However, this conclusion is not novel.

We are not the first ones to report the strong impact of growth rate on promoter activity, and we explicitly refer to previous reports showing similar effects in the introduction and discussion of our manuscript. However, we maintain that our straightforward computational approach to dissect global and specific regulation is indeed novel. We emphasized this aspect in the revised manuscript (L192-193).

4. *My major argument concerning the novelty and originality of the manuscript was based on extrapolation of previous work on global regulation and a MODEL (as I said) showing that four sensors are sufficient to explain the metabolic adaptation during a diauxic shift from glucose to acetate. I fully agree that the Kotte publication is modeling work. Nevertheless, some experimental evidence the authors are certainly aware of has been added later. The first author of the present manuscript is also first author of the PNAS publication showing experimental evidence for one of the flux sensors, FBP.*

I respectfully apologize and admit that my initial judgment may have been a little harsh. The conclusions of the present work were expected (in my opinion). The major merit of the manuscript is to EXPERIMENTALLY SHOW that this expectation was warranted. I agree with this claim.

We agree that a key point of our manuscript is to show experimentally that few regulatory metabolites are sufficient to explain most of the observed specific transcriptional regulation. We apologize if this point was not clear enough in the initial submission.

5. *OK*

In summary: The revised manuscript is improved in many respects. The conclusions are not surprising. The major virtue of the work is to experimentally demonstrate that two transcription factors (and essentially two metabolites, considering FBP and FIP as closely related) control most of the specific expression of genes involved in central carbon metabolism. Since the major strength of the work is experimental, all of the original, raw data (and error estimates) should be provided in supplementary files.

We thank the reviewer for the positive assessment of our revised manuscript, and thank all reviewers for their comments. Your feedback has helped a lot in improving this manuscript. We included all data (with error estimates) as supplementary files (EV tables 1 to 8).

Corresponding Author Name: Uwe Sauer

Manuscript Number: MSB-16-7141